# Moisture adsorption-desorption full cycle power generation

Haiyan Wang[1], Tiancheng He[2], Xuanzhang Hao[2], Yaxin Huang[2], Houze Yao[2], Feng Liu [3✉], Huhu Cheng [1✉] & Liangti Qu [1,2✉]

Environment-adaptive power generation can play an important role in next-generation energy conversion. Herein, we propose a moisture adsorption-desorption power generator (MADG) based on porous ionizable assembly, which spontaneously adsorbs moisture at high RH and desorbs moisture at low RH, thus leading to cyclic electric output. A MADG unit can generate a high voltage of ~0.5 V and a current of 100 μA at 100% relative humidity (RH), delivers an electric output (~0.5 V and ~50 μA) at 15 ± 5% RH, and offers a maximum output power density approaching to 120 mW m$^{-2}$. Such MADG devices could conduct enough power to illuminate a road lamp in outdoor application and directly drive electrochemical process. This work affords a closed-loop pathway for versatile moisture-based energy conversion.

---

[1] Key Laboratory of Organic Optoelectronics & Molecular Engineering, Ministry of Education, Department of Chemistry, Tsinghua University, Beijing 100084, China. [2] State Key Laboratory of Tribology, Department of Mechanical Engineering, Tsinghua University, Beijing 100084, China. [3] State Key Laboratory of Nonlinear Mechanics, Institute of Mechanics, Chinese Academy of Sciences, Beijing 100190, China. ✉email: liufeng@imech.ac.cn; huhucheng@tsinghua.edu.cn; lqu@mail.tsinghua.edu.cn

As recyclable resource, water is not only vital to life but also represents the largest energy carrier, regulator and balancer on the Earth, predominantly supplied through hydrologic cycle[1,2]. The omnipresent hydrologic cycle involves the transformation from liquid to gaseous water during evaporation, such as evaporation of oceans, and the reverse during condensation, such as precipitation in clouds, which provides tremendous energy exchange[3,4]. Such underlying energy evolving into a rich variety of forms, approaches to around $60 \times 10^{15}$ W per year, several orders of magnitude over the average consumption of human activities by electricity, but very little of these energy has yet been harnessed[5–7]. Recently, moisture-enabled electricity generation technology has been developed to meet the energy demand in isolated off-grid areas[8–11], which viably leverage the adsorption interaction with atmospheric water molecules by hygroscopic materials, converting the chemical energy from moisture into electric power[12,13]. However, moist-electric generator (MEG) will discontinue delivering electric output as the single process of water adsorption tends to an equilibrium state[14–17], reflecting its bottleneck of unsustainable and non-repetitive power generation. Alternative variation between high relative humidity (RH) and low RH in a day is a common natural phenomenon along with hydrologic cycle (Supplementary Fig. 1). Owing to the heavy dependence between high RH and electricity-generating performance of MEG, it also remains a challenge to generate power according to RH changes in the dynamic environment.

In this work, we propose a moisture adsorption-desorption power generator (MADG) by adopting three-dimensional (3D) porous and ionizable assembly with surrounding encapsulation. The MADG not only exerts moisture adsorption power generation at high RH but also endows moisture desorption electricity generation at low RH based on ions diffusion, dominated by ions concentration difference and ion-hydration energy, respectively (Fig. 1). This postulated working mechanism is further reasonably verified by solid-state nuclear magnetic resonance (NMR) experimental tests combining with theoretical calculations. Compared with previous MEG by harnessing a single adsorption process, the full cycle MADG integrates adsorption and desorption enabled power generation into a closed-loop process, thus it could afford repeatable electricity-generating performance, as well as convert versatile moisture-based energy into electricity. A MADG unit can generate a high voltage of ~0.5 V and a current of ~100 μA at 100% RH (water adsorption), and deliver electric output (~0.5 V and ~50 μA) at 15 ± 5% RH (water desorption). A maximum output power density approaches to 120 mW m$^{-2}$ in MADG, realizing a superb trade-off between internal resistance and maximum output power density for contribution to direct external power supply. Correspondingly, the MADGs could directly supply enough power to drive commercial electronic devices and electrochemical process for a long term, as well as perform continuously full cyclic power generation according to dynamic RH under practical outdoors.

## Results

**Fabrication and characterization of electricity-generating materials.** The electricity-generating film consists of sodium alginate (SA), silicon dioxide nanofiber (SiO$_2$), and reduced graphene oxide (rGO), denoted as SAG film (Fig. 2a). Briefly, the natural product of SA featured with abundant carboxyl functional group (e.g., -COONa), plays the role in dissociating mobile Na$^+$ ions in response to interaction with water molecules. The SiO$_2$ nanofiber is beneficial to construct hierarchical pore structure, facilitating water molecules and ions transport coupled with mechanic stability in water[18,19]. And the rGO, as a two-dimensional (2D) nanosheet, is employed to assemble 3D conductive skeleton and tune resistance[20]. The flexible SAG film with

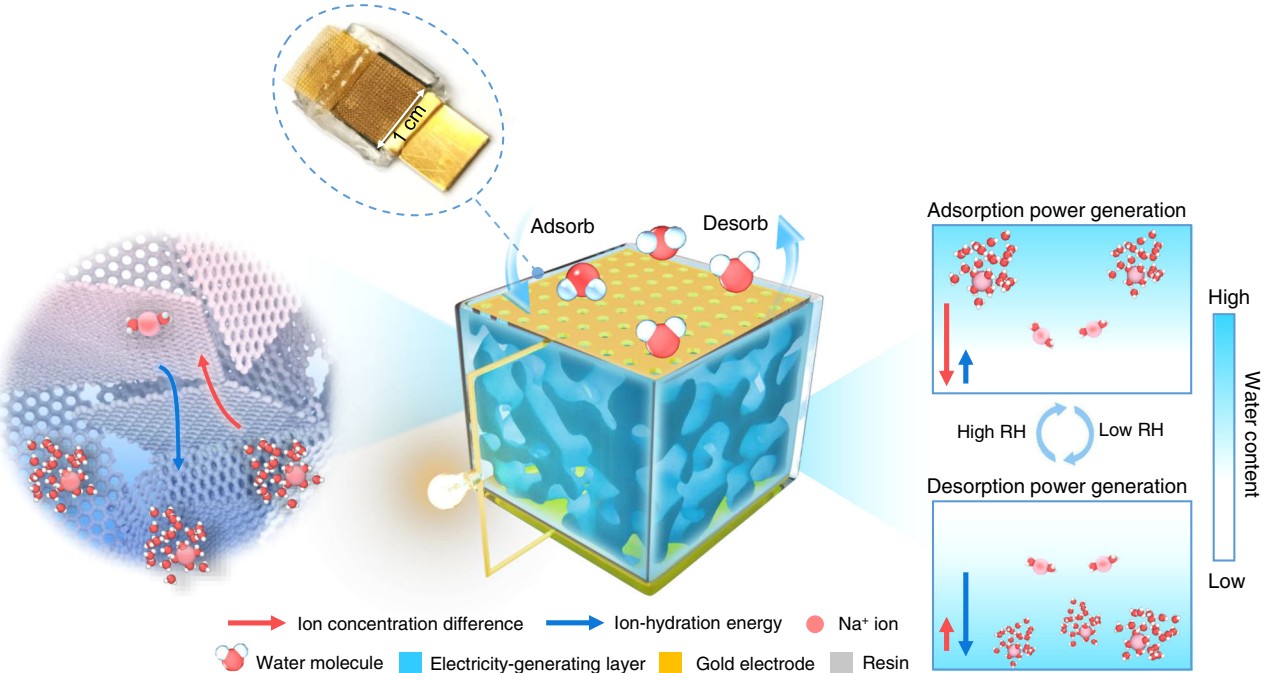

**Fig. 1 Design of MADG.** Scheme of the device structure, working principle, and photograph of device. The MADG is composed of gold electrodes, 3D ionizable porous assembly as electricity-generating material, and encapsulation layer, where the upper gold electrode has holes to allow entry/removal of moisture. Benefiting from well-designed structure, the MADG enables to deliver electric output under high and low RH condition through asymmetric moisture adsorption and desorption process, respectively. The electricity-generating principle is attributed to hydrated ions diffusion, driven by ion concentration difference during moisture adsorption power generation and dominated by ion-hydration energy during moisture desorption electricity generation, respectively.

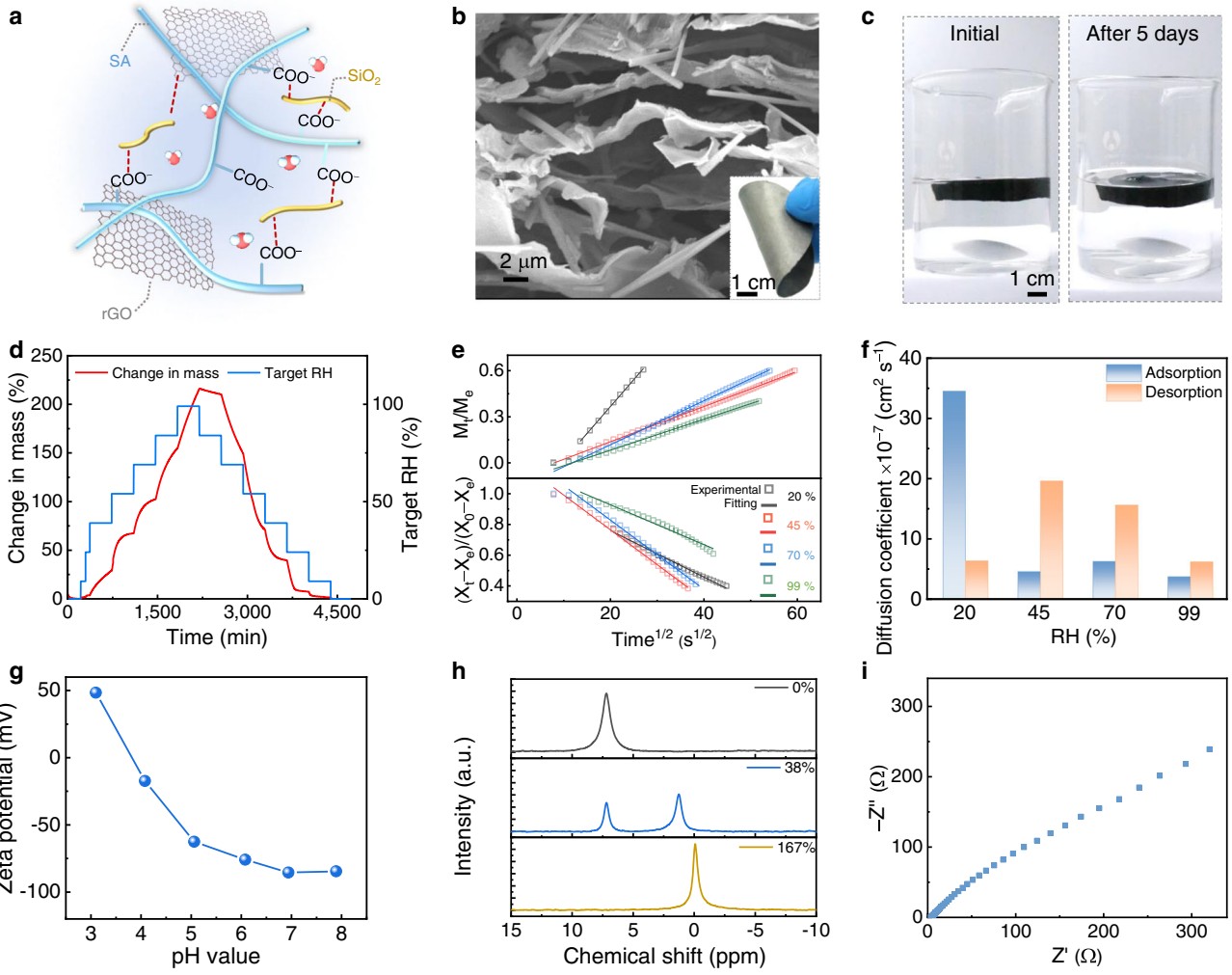

**Fig. 2 Structure and properties of SAG film. a** Schematic of the SAG film composed of SA chains, SiO$_2$ nanofiber, and rGO nanosheet. **b** Scanning electron microscopic (SEM) image of SAG film. Inset is the photograph of flexible SAG film. **c** Photos of SAG film in water during stirring (200 rpm) before and after 5 days. **d** Moisture adsorption and desorption kinetics for SAG film at 40ºC. **e** $M_t/M_e$ and $(X_t-X_e)/(X_0-X_e)$ as a function of square root of time.
**f** Adsorption and desorption diffusion coefficient of SAG film under different RH at 40ºC. **g** Zeta potential of SAG film within the pH range from 3 to 8.
**h** $^{23}$Na solid state NMR spectra of SAG film with various water content. **i** The electrochemical impedance spectra of SAG film at 100% RH.

porous structure was easily fabricated by using freeze-drying and chemical reduction method (Fig. 2b and Supplementary Figs. 2–4). The rGO nano-sheets are uniformly coated with the SA chains and SiO$_2$ nanofiber with diameter of ~500 nm (Supplementary Figs. 5 and 6). Moreover, the SAG component is highly stable in the water (Fig. 2c), further supported by negligible change of mechanical and morphologic properties of SAG film before and after in water (Supplementary Fig. 7). Water sorption kinetics and isotherm profiles displays the SAG film enables to afford water-adsorbing capacity of up to 210% under testing condition of ~100% RH and 40ºC (Fig. 2d and Supplementary Fig. 8), indicating its super hydrophilicity. The moisture diffusion coefficient ($D$), reflecting the rate of water transfer, can be determined from Fick's second law in the form of a trigonometric series as following[21,22]:

$$\frac{M_t}{M_e} = \frac{2}{d}\sqrt{\frac{Dt}{\pi}} \left(0 < \frac{M_t}{M_e} < 0.6\right) \quad (1)$$

$$\frac{X_t - X_e}{X_0 - X_e} = 1 - \frac{2}{d}\sqrt{\frac{Dt}{\pi}} \left(0.4 < \frac{X_t - X_e}{X_0 - X_e} < 1\right) \quad (2)$$

where $M_t$ and $M_e$ represents the mass of water uptake at the time $t$ and equilibrium for adsorption process, $X_0$, $X_t$, and $X_e$, is defined as the mass of water uptake at time 0, $t$, and equilibrium for desorption process, $d$ is the sample thickness. Experimental values of $M_t/M_e$ or $(X_t-X_e)/(X_0-X_e)$ are plotted as a function of the square root of time at various RH, along with the linear fit (Fig. 2e, Supplementary Fig. 9 and Table 1). The adsorption and desorption diffusion coefficient of SAG film at different RH and 40ºC is summarized as Fig. 2f, which approaches to $3.8 \times 10^{-7}$ and $6.3 \times 10^{-7}$ cm$^2$s$^{-1}$ at 99% RH, respectively, confirming the rapid water transfer in SAG film.

The SAG film is characterized with carboxyl functional group, as verified by Fourier transform infrared and X-ray photoelectron spectroscopy results (Supplementary Fig. 10). Zeta potential testing presents the SAG film is negatively charged following functional group dissociation (Fig. 2g), which is consistent with the surface potential of SAG film (Supplementary Fig. 11). Furthermore, the content of dissociated Na$^+$ ions of SAG film is investigated by solid-state NMR testing (Fig. 2h). Such chemical shift of[23] Na visibly deviated to the right with increased water content of SAG film, approaching to a chemical shift of ~0 ppm with the water content of 167%, coinciding with that of

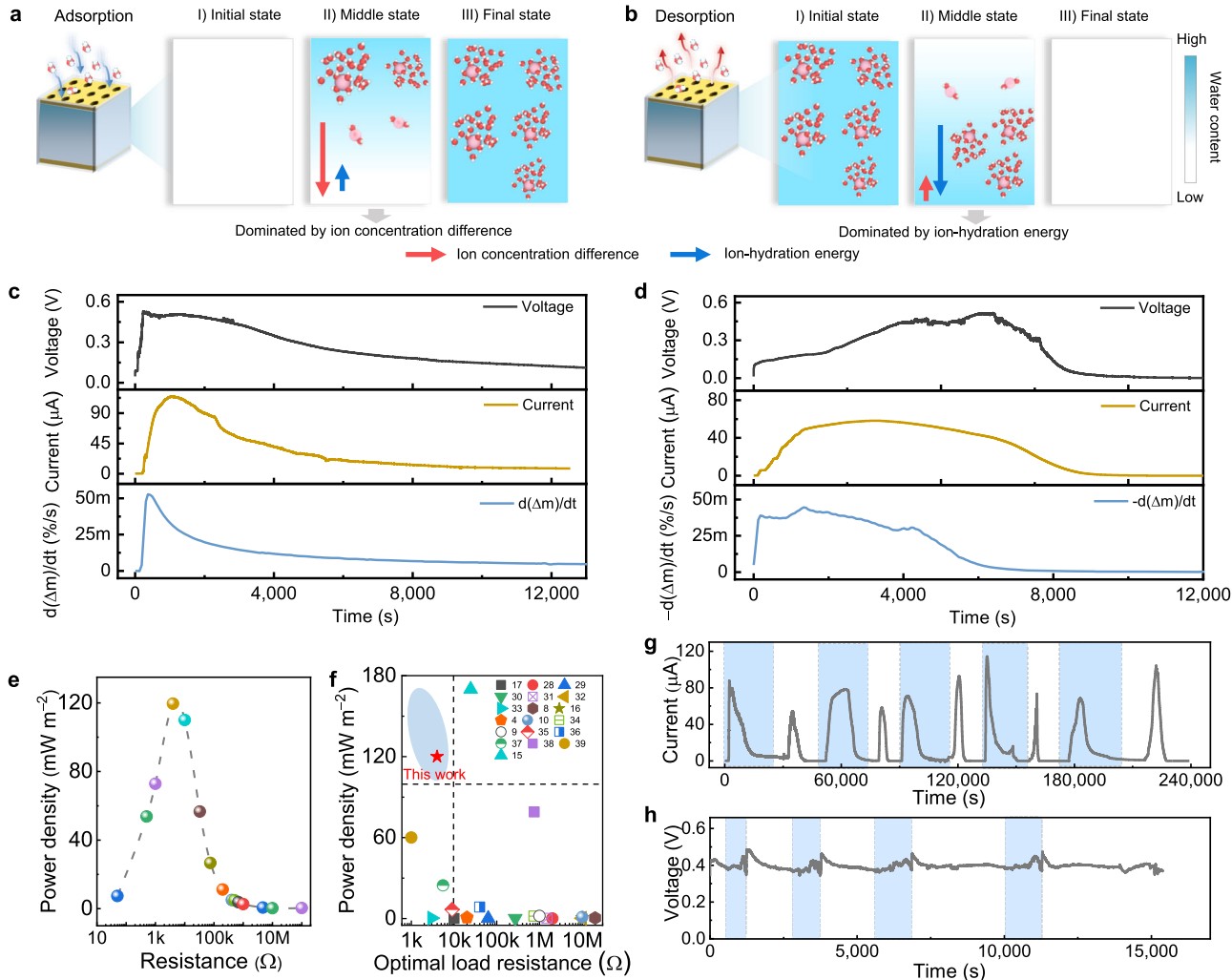

**Fig. 3 Electricity generation of a MADG unit.** Schematic illustrations of the postulated working principle of **a** moisture adsorption power generation and **b** desorption power generation. **c** Open-circuit voltage, short-circuit current, rate of adsorbing water molecules ($d(\Delta m)/dt$) during moisture-adsorbing electricity generation under ~100% RH and 40°C. **d** Open-circuit voltage, short-circuit current, rate of desorbing water molecules ($-d(\Delta m)/dt$) during moisture-desorbing energy generation at ~15 ± 5% RH and 40°C. **e** Output power density as a function with varied electric resistances. **f** Comparison of output power density and internal resistance between MADG and reported moisture induced generators. **g** Cyclic current output of repeatable moisture adsorption and desorption power generation for MADG in a simulated dynamic ambient condition. The blue region represents adsorption process at high RH condition (90 ± 10%, 30 ± 10°C) and white region denotes desorption process at low RH environment (15 ± 5%, 30 ± 10°C). The switch point of two processes are selected at finial state of adsorption and desorption. **h** Voltage output of MADG device in response to switchable RH. The switch point is selected at middle state of adsorption and desorption.

completely mobile Na⁺ ions of an aqueous sodium sulfate solution (Supplementary Fig. 12). Thus, the Na element of SAG film totally exists in the form of mobile Na⁺ ions with water uptake of 167%, and the content of dissociated Na⁺ ions is up to $4.8 \times 10^{-3}$ mol kg⁻¹ measured by calibration curve method[23] (Supplementary Figs. 13 and 14), implying a high-efficient dissociation ability of SAG film. Benefiting from interconnected 3D skeleton structure and abundant mobile ions, the SAG film exerts great ionic transportation along with nano-structures, and affords the ion conductivity of 0.11 S m⁻¹ (Fig. 2i and Supplementary Fig. 15).

**Electricity-generating performance of MADG.** The MADG unit is composed of SAG film as electricity-generating layer sandwiched between a pair of gold electrodes, and sealed with resin (Fig. 1). This encapsulation layer not only ensures asymmetric adsorption and desorption of water molecules, but also exerts

confinement function for SAG film. The electricity generation involves two processes: moisture adsorption power generation driving by ion concentration difference and desorption power generation dominated by ion-hydration energy. During moisture adsorption power generation at high RH (Fig. 3a), moisture gradually increases from top to bottom in the device, which leads to asymmetrical moisture adsorption and Na⁺ ions dissociation, accordingly forming ion concentration difference in the MADG device. And the negatively charged groups anchored on the backbone of polymer chains remain immovable. It could be seen that nearly no hydrated dissociated Na⁺ ions exist in the lower regions at the initial and middle states of adsorption, as a result diffusion can only drive hydrated Na⁺ ions from upper region to lower region driven by ion concentration difference, thus leading to an electric output[9,15]. With the adsorption saturation of device, ions will tend to be uniformly distributed in the final state. Subsequently, the saturated MADG can conduct moisture desorption power generation (Fig. 3b). Already dissociated hydrated

Na$^+$ ion would be surrounded by less water molecules with decreasing water content in MADG, while it still remains hydrated state rather than direct variation to bonded state (middle state in Fig. 3b). Thus, there is lack of effective ion concentration difference served as driving force in the system. Due to asymmetrically desorption, the water content descends from upper region to lower region in system, thus causing that hydrated Na$^+$ ion in upper region will be surrounded by fewer water molecules than those in lower region. And the hydrated Na$^+$ ion combined with more water molecules would possess the lower ion-hydration energy, attributed to the formation of stable configuration. Thus, the hydrated Na$^+$ ions tend to move from upper region to lower region dominated by ion-hydration energy[24,25], inducing the formation of net ions flow and electric signal.

The MADG can generate an open-circuit voltage of up to 0.5 V and a short-circuit current approaching to 100 μA at 100% RH by directionally adsorbing water molecules from atmosphere (Fig. 3c and Supplementary Fig. 16). The hydrated MADG spontaneously desorbs water molecules under low RH (15 ± 5% RH), rendering an open-circuit voltage of ~0.5 V and short-circuit current of ~50 μA (Fig. 3d). The current-voltage (I-V) curves further identify the electricity-generating performance of MADG (Supplementary Fig. 17). More importantly, the variation of hydration and dehydration rate is coincident with that of electric output (Supplementary Fig. 18), implying that electricity generation is originally induced by moisture adsorption and desorption.

The MADG could offer electric power for external load. From Ohm law of whole circuit, the output voltage of V$_{load}$ and power density of P$_{load}$ across load is calculated as:[26]

$$V_{load} = \frac{V_{oc} R_{load}}{R_0 + R_{load}} \quad (3)$$

$$P_{load} = \frac{(V_{load})^2}{s R_{load}} = \frac{(V_{oc})^2 R_{load}}{s (R_0 + R_{load})^2} \quad (4)$$

where V$_{oc}$, R$_{load}$, R$_0$, $s$ represents the open-circuit voltage, load resistance, internal resistance of MADG, area of electricity-generating material, respectively. When R$_{load}$ is equal to R$_0$, P$_{load}$ will reach to maximum values, which is considered as matched working-status for MADG[27]. Thus, the parameters of internal resistance and output power density are both great significance for low-loss energy supply of generators. The MADG can effectively offer electric output and exert a maximum output power density of 120 mW m$^{-2}$ at an optimal resistance of 4 kΩ (Fig. 3e, Supplementary Figs. 19 and 20). The load resistance of sorts of small commercial electronics might generally lie in the range of below 1 kΩ (Supplementary Table 2). Thus, the desirable internal resistance of generator needs to match electronics resistances as close as possible, to ensure directly high-efficient power supply for small electronics. Notably, the MADG has a good trade-off between internal resistance and maximum areal or volumetric output power density, prior to that of reported MEGs[4,8–10,15–17,28–39], facilitating a high working output (Fig. 3f, Supplementary Fig. 21, Table 3 and 4).

Given that the fluctuation of ambient environment, we further simulated a dynamic ambient condition of in lab to examine the electricity-generating performance. Fig. 3g, h displays the continuous electric output of MADG at a switchable RH of 90 ± 10% and 15 ± 5%, demonstrating that the MADG device possesses environment-adaptive cyclic performance. After the multiple cycles of test, the chemical composition of SAG film is almost unchanged (Supplementary Fig. 22), which is critical for repeatable power generation. In comparison to moisture-enabled generators based on single adsorption process, the MADG is no longer limited by the bottlenecks such as adsorption equilibrium and sever dependence on high RH. This is attributed to concurrently utilize adsorption and desorption process. In this regard, the MADG is featured with environment-adaptive ability and cyclic power generation, which could favor in practical applications.

**Optimization of performance**. The electricity-generating performance based on single or hybrid component is shown in Fig. 4a and Supplementary Fig. 23. The SiO$_2$, rGO, or SiO$_2$/rGO film is unable to exert energy generation due to the absence of mobile ions dissociated form materials. And the SA or SA/SiO$_2$ film yields a low current of below 1 μA and 7 μA, respectively, resulting from tremendous internal resistance. Based on synergistic effect, the interconnected SAG film with high porosity of 77% can possess a striking water-adsorbing capacity, diffusion coefficient of water molecules, zeta potential and ionic conductivity (Fig. 4b, Supplementary Figs. 24–27), reflecting its excellent water molecules transport, ions dissociation and diffusion. Correspondingly, the hybrid SAG film renders a considerable voltage and current output, outperforming other components.

Owing to the effect of SAG thickness on ions concentration gradient and diffusion[8,14] (Supplementary Fig. 28), the electric signals exhibited ascending and then descending variation as the increasing thickness (Fig. 4c). Chemical reduction temperature could tune the reduction degree of rGO component, making the variation of resistance and hydrophilicity of SAG film (Supplementary Fig. 29). The SAG film with reduction temperature of 80°C, generated an optimal performance (Fig. 4d). The compressing strength in the tableting process could regulate the bulk density of SAG film, further regulating resistance and ions transport (Supplementary Fig. 30)[10]. As a result, the maximum output was provided when the SAG was compressed with 10 MPa (Fig. 4e). Besides, both externally environmental RH and temperature have an effect on the performance of MADG device, directly associated with sorption kinetics and ions diffusion (Supplementary Figs. 31, 33). The device assembled with other inert electrodes also delivers similar performance (Supplementary Fig. 34). Expectedly, analogous materials served as electricity-generating layer, and viably yields a voltage and current output, confirming the universality of moist-adsorb-desorb-generation (Fig. 4f and Supplementary Fig. 35).

**Verification for electricity-generating mechanism**. The power generation of MADG is attributed to ions diffusion, driven by ion concentration difference during moisture adsorption power generation and dominated by ion-hydration energy during moisture desorption electricity generation, respectively (Fig. 3a and b). The driving force of ion concentration difference could prevail for adsorption process at high RH condition, which is well consistent with working mechanism of previous works about moisture adsorption power generation[8–11]. The ion diffusion dominated by ion-hydration energy is firstly proposed for moisture desorption power generation, which mainly relies on two points, including: (1) Hydrated Na$^+$ ions number remains unchanged but only surrounding water molecules number reduces with decreasing water content within a wide range, thus there is lack of effective ion concentration difference served as driving force in the system; (2) Hydrated Na$^+$ ion combined with more water molecules would possess lower ion-hydration energy than that of hydrated Na$^+$ ion surrounded less water molecules.

To verify the first point, we have conducted solid state NMR testing to measure the variation of[23]Na chemical shift of SAG film with different water content. As shown in Fig. 5a, two peaks of chemical shift correspond to boned Na atom anchored on the

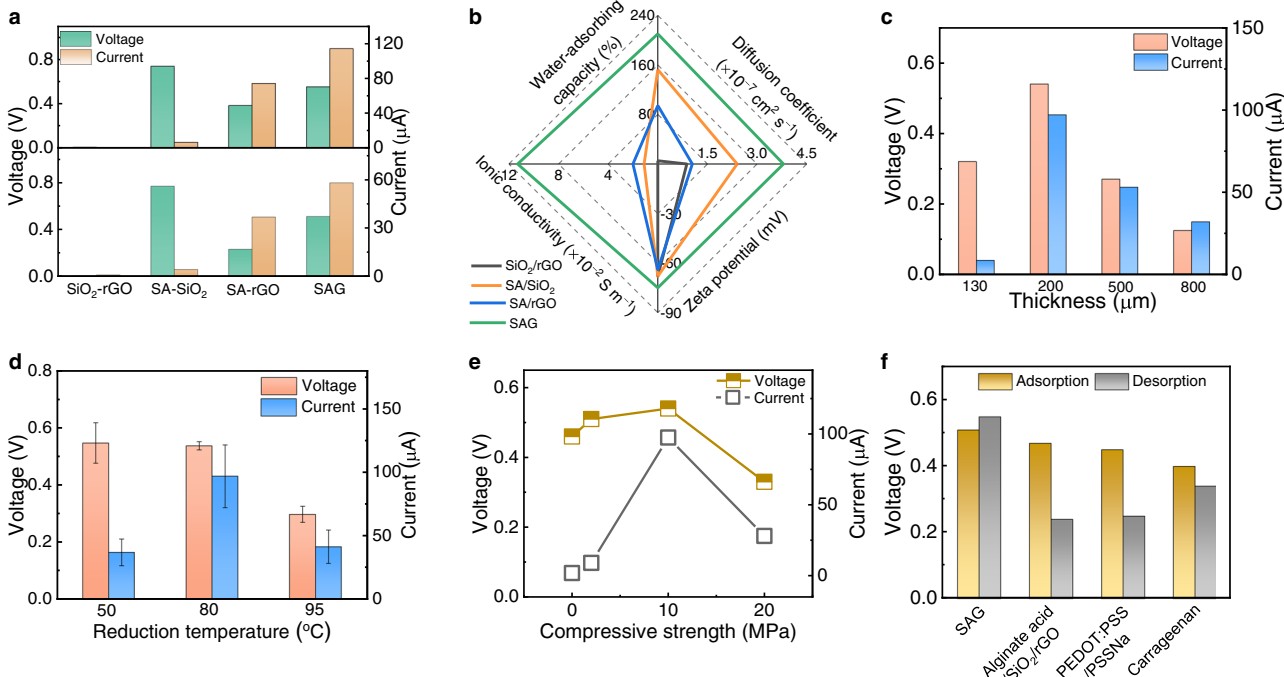

**Fig. 4 Factors that influence the electricity-generating performance of MADG. a** The electric output of MADGs based on SiO$_2$/rGO, SA/SiO$_2$, SA/rGO, and SAG film. **b** Radial plot comparing the water-adsorbing capacity, diffusion coefficient of water molecules, zeta potential at pH~7, and conductivity for SiO$_2$/rGO, SA/SiO$_2$, SA/rGO, and SAG film. Voltage and current output based on SAG film with **c** different thickness, **d** reduction temperature and **e** compressive strength at 90 ± 10% RH and 35 ± 5 °C. Error bars represent the standard deviations from multiple measurements. **f** Electric output offered by MADGs based on other analogous materials, including alginate acid/SiO$_2$/rGO, PEDOT: PSS/PSSNa (that is, sodium polystyrene sulfonate), and carrageenan film.

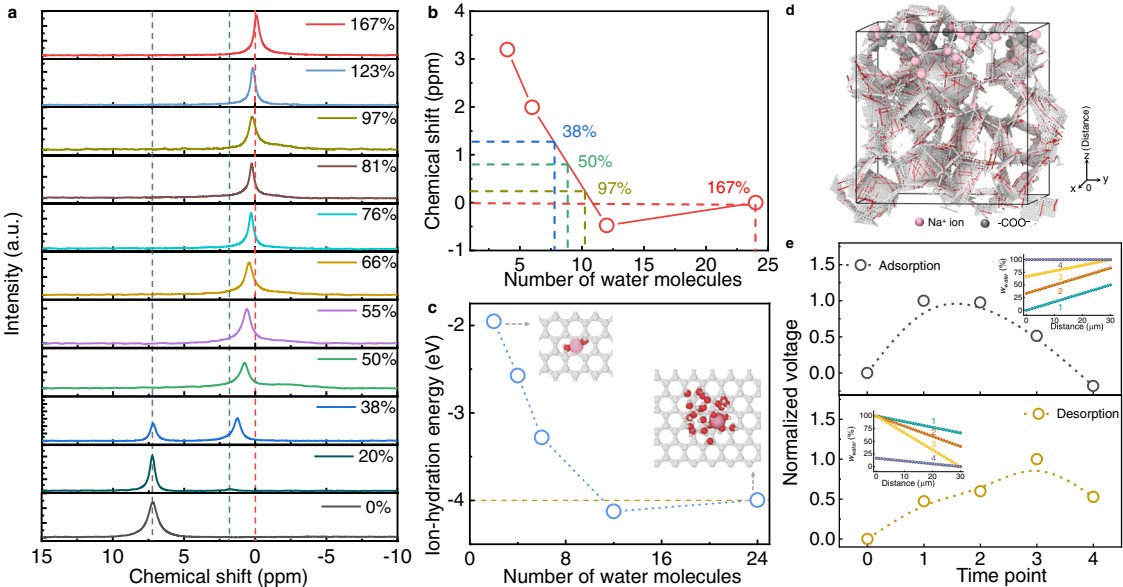

**Fig. 5 Verification of working mechanism by NMR experiments and theoretical calculations. a** $^{23}$Na solid NMR spectra of SAG film with different water content during desorption process. **b** Chemical shifts variation with number of water molecules according to first principle calculation. **c** Simulated ion-hydration energy of hydrated Na$^+$ ion with different numbers of water molecules by MD simulations. Insets are the calculated configurations of hydrated Na$^+$ ion with 2 or 24 water molecules. Hydration enthalpy of Na$^+$ ions is about −4 eV in aqueous solution. **d** Model of kinetic Monte Carlo simulation. The positively charged ions of Na$^+$ and negatively charged groups anchored on the backbone could be ionized by interaction with water molecules in theoretical model. **e** The calculated normalized voltage under varied water content distribution during adsorption and desorption process. Time point represents a moment of adsorption or desorption process. Insets are water content distribution along with distance under different time point. W$_{water}$ denotes as water content.

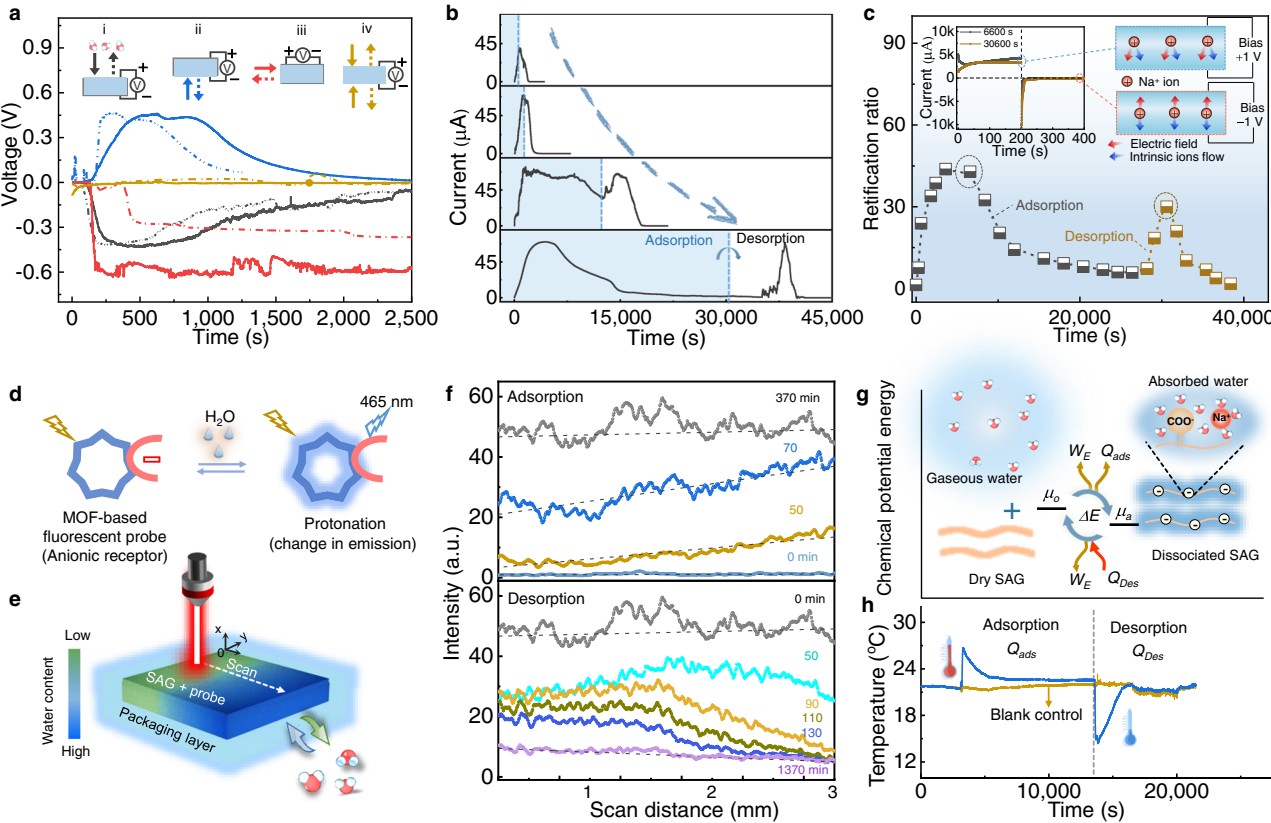

**Fig. 6 Demonstration of working mechanism and discussion of energy sources. a** Electric signal of MADG in response to moisture sources in different directions. Insets are schematic illustrations of four directions of moisture source interacted with MADGs. **b** The current-time profile in the electricity generation with various adsorption time. **c** The variation of ions rectification ratio of MADG under an alternating bias of ±1 V during adsorption and desorption process (40 °C). Insets are current response curves and the schematic diagram of ions transport. **d** Scheme of the mechanism of water detection by fluorescence sensing. The probe is a type of luminescent metal-organic framework. A trace of water is absorbed in the pores of metal-organic framework-based probe, resulting in its protonation and turn-on blue fluorescence (465 nm). **e** Illustration of experimental set-up of CLSM testing. **f** Distribution of fluorescence intensity along with scanning direction during adsorption and desorption. **g** Schematic illustration of proposed energy input and conversion involved in MADG. The $\mu_o$ and $\mu_a$ represents the total chemical potential energy of both gaseous water molecules and dry SAG film, as well as dissociated SAG film with adsorbed water, respectively. The $\Delta E$, $W_E$, $Q_{Ads}$, and $Q_{Des}$ denotes as the variation of chemical potential energy, electricity energy, heat of adsorption, and heat of desorption. **h** The variation of temperature of SAG film during the moisture adsorption and desorption process.

polymer chains and dissociated $Na^+$ ions with hydrated state, respectively. The bonded Na stably gives rise to a peak at the chemical shift of 7.2 ppm (marked by black dash line in Fig. 5a), while the chemical shift of hydrated $Na^+$ ion visibly deviated from 0 to 1.7 ppm with decreasing water content from 167% to 20% (marked by green and red dash line) with water content. More interestingly, when the water content gradually decreases from 167% to 50%, NMR spectra always present the only peak of chemical shift of dissociated $Na^+$ ion coupled with a gradual positive shift based on shielding effect, demonstrating that $Na^+$ ion can keep dissociated hydrated state in such wide range (167% ~50%) but surrounding water molecules will be reduced. Furthermore, based on first principle calculation, the number of water molecules around $Na^+$ ion corresponding to chemical shif could be quantitative (more calculation details could be found in Supplementary Information). Fig. 5b indicates the hydrated $Na^+$ ions will be surrounded by less number of water molecules with reduced water content, reconfirming the first point mentioned above. To further prove the second point, the hydration energy of hydrated $Na^+$ ion is calculated by moleculer dynamic (MD) simulations. As displayed in Fig. 5c, the hydrated ion combined with more water molecules would possess the lower ion-hydration energy, attributed to the formation of stable configuration. Therefore, it is reasonably supported that ion-hydration

energy could be serve as a driving force. Moreover, postulated working principle is theoretically supported by kinetic Monte Carlo simulation (Fig. 5d). As displayed in Fig. 5e, the water content distribution of time point 1, 2, 3, and 4 could be compared with 50, 60, 70, and 370 min in adsorption process, and 50, 70, 90, and 130 min in desorption process (see Fig. 6f). The calculated normalized voltage under varied water content distribution is close with experimental results.

Meanwhile, related electric evidences and visualization of water content distribution are obtained for confirming the proposed mechanism. First, electric signals generated by adsorption and desorption power generation perform the same direction (Fig. 6a), demonstrating the directions of charged ions diffusion are the same for two processes. Thus, there is certainly other driving force for the desorption power generation, according with above proposed mechanism. Second, the moisture desorption power generation was observed under different adsorbed states (Fig. 6b). The curves of moisture-adsorbing and moisture-desorbing power generation, exhibit approximately symmetrical shape, reflecting analogous ions diffusion in such two processes. Third, by employing external electric field perturbation on MADG, its real-time ion rectification is in situ recorded during moisture adsorption and desorption generation (Fig. 6c, Supplementary Figs. 36 and 37). The net ions transport could be adjusted by both

applying external electric field and intrinsic ions flow dominated by ions concentration difference or ion-hydration energy[40]. Thus, the synergy between two fields can improve current value, otherwise the opposite, leading to ion rectification. The MADG presents a similar variation of rectification ratio during moisture adsorption and desorption power generation with an identically alternating bias of ±1 V, and displays maximum rectification ratio of 44 during adsorption and 30 during desorption, reconfirming that MADG yields spontaneous ions flow in the same direction during adsorption and desorption process. Finally, we adopted fluorescence sensing method to visualize the distribution of water content in SAG film during asymmetric water adsorption/desorption by confocal laser scanning microscope (CLSM) testing. This probe of the hybrid of tris(2,2'-bipyridyl) ruthenium (II) and metal−organic framework[41], interacted with water molecules, could provide emission at 465 nm under ultraviolet excitation based on the protonation of the nitrogen atom (Fig. 6d and Supplementary Fig. 38). The SAG film infused with probe is reasonably packaged for real-time CLSM testing, which ensures directional adsorption or desorption of water molecules from one side (Fig. 6e). The original sample is unable to yield the fluorescence intensity before adsorbing moisture. Then the fluorescence intensity visibly enhances along with the scanning direction of laser with increasing adsorption time, illustrating the existence of water content gradient in the device during adsorption (Fig. 6f and Supplementary Fig. 39). Once the adsorption reaches equilibrium, the fluorescence intensity displays almost uniform distribution. When hydrated device conducts desorption, the fluorescence intensity shows slightly changed and then obviously decreased along with the scanning direction (Supplementary Fig. 40).

In the spontaneous adsorption process, the variation of chemical potential energy, reasonably served as energy source[7,42], could convert into electricity energy and heat of adsorption (Fig. 6g, h). On the other hand, the MADG device enables to draw the thermal energy from ambient environment by converting the sensible heat into latent heat during desorption process[1], leading to the generation of electric power and the variation of chemical potential energy. Thereupon the energy source is green and renewable for electricity generation of MADG.

**Applications of MADG.** The integration of units is essential to scale up the energy output of device. The generated voltage is up to about 11 V by connecting 21 units in series, increasing linearly with an average of 0.53 V per unit (Fig. 7a and Supplementary Fig. 41a). The current output of 16 units in parallel connection is boosted to about 1.3 mA, demonstrating a scaling performance of MADG (Fig. 7b and Supplementary Fig. 41b). The electric power supplied by MADGs is capable of charging commercial capacitors of 0.47, 47 or 470 mF up to 3 V using an integrated 6 × 3 arrays (6 series and 3 parallel connection, Fig. 7c). Due to poor power-generating performance and immensely internal resistance, previously reported generators induced by moisture could only drive commercial electronics for short bursts or by using additional capacitors[8–10], which severely impedes their practical applications. Benefiting from exceptional output power density and desirable internal resistance, the MADGs as a power source enable to directly power commercial electronics for long time without the requirement for any extra rectifying circuits and energy-storage devices. For instance, 6 × 6 integrated device is sufficient to continuously illuminate 6 red light-emitting diodes (LEDs) and a white LED for 6 h and 3 h under one electricity-generation cycle, respectively (Fig. 7d, Supplementary Fig. 42 and Movie 1).

Also, the electric energy generated by MADGs is employed to directly drive an electrochemical polymerization and electrochromic of polyaniline (PANI, Supplementary Fig. 43). For self-powered electro-polymerization, the MADG served as energy source is connected with electrochemical cell (Supplementary Fig. 44), as well as platinum and fluorine doped tin oxide (FTO) glass covered with mask layer are used as electrodes. Aniline monomer could be simply polymerized into arbitrarily patterned PANI film assembled by nanoparticles, such as "THU" and dimensional barcode patterns (Supplementary Figs. 45−47), reconfirmed by element mapping and cyclic voltammetry (Supplementary Figs. 48−50). Above PANI film is further used as active material in self-powered electrochromic system, where MADG provide working voltage and current (Supplementary Fig. 51 and 52). Based on reverse switch between oxidized/reduced state[43], the color of PANI film enables to exchange between blue and green (Fig. 7e, f, Supplementary Fig. 53), further suggesting its application in self-powered electrochemical-related devices.

In order to demonstrate the practical applicability, the electric output of MADG has been investigated in the natural condition. As schemed in Fig. 7g, the MADG tends to perform moist-adsorb generation in the region closed to water surface with ~80−100% RH, while the device is prone to yield moist-desorb generation far away from water area with ~20−40% RH. Attached to a rotatable board, the MADG device could intelligently self-transforms power-generating manner and delivers a continuous voltage output of ~0.5−0.6 V (Fig. 7h), elucidating its applicability under dynamic humidity environment. In the outdoor test at Beijing, the generated voltage slightly fluctuates at 0.46 V for about 4 days under a waving environment of ~25%−90% RH and 15°C−30°C (Fig. 7i). In addition, a proof-of-concept self-powered road lamp is successfully designed, which is composed of a power source of MADG, a booster circuit for voltage management, and a capacitor for electricity storage (Fig. 7j). The continuously electric output of MADG can be stored into capacitor, then accumulated electric energy viably power street lamp at light via a booster circuit on demand, exhibiting the outdoor utility of MADG.

## Discussion

In summary, we developed MADG device by adopting synergistic SAG film and specific package. A MADG unit can produce open-circuit voltage of ~0.5 V and a short-circuit current of ~100 μA with a maximum output power density of up to 120 mW m$^{-2}$. Based on directional moisture transport and ions diffusion, the MADG integrate moisture adsorption power generation driven by ion concentration and desorption power generation dominated by ion-hydration energy, endowed with cyclic electric output and environment-adaptive ability. Furthermore, experimental tests combining with theoretical calculations provide strong evidences to the existence of ion-hydration energy driving force. The supplied energy of MADG directly powers commercial electronics for long term, and creatively drives electrochemical processes, as well as conducts electricity generation in response to varied RH in practical environment. Therefore, MADG provides a powerful approach for sustainable energy harvesting and conversion. For its future exploration, the reasonable regulation of electricity-generating materials (such as abundant functional group, hierarchically porous and robust structure, and stability) and device structure design (such as encapsulation layer, one-all-in structure, and circuit design), will be potential for achieving efficient and repeatable moisture adsorption-desorption and fast ion diffusion, enabling to further improving the performance of MADG device.

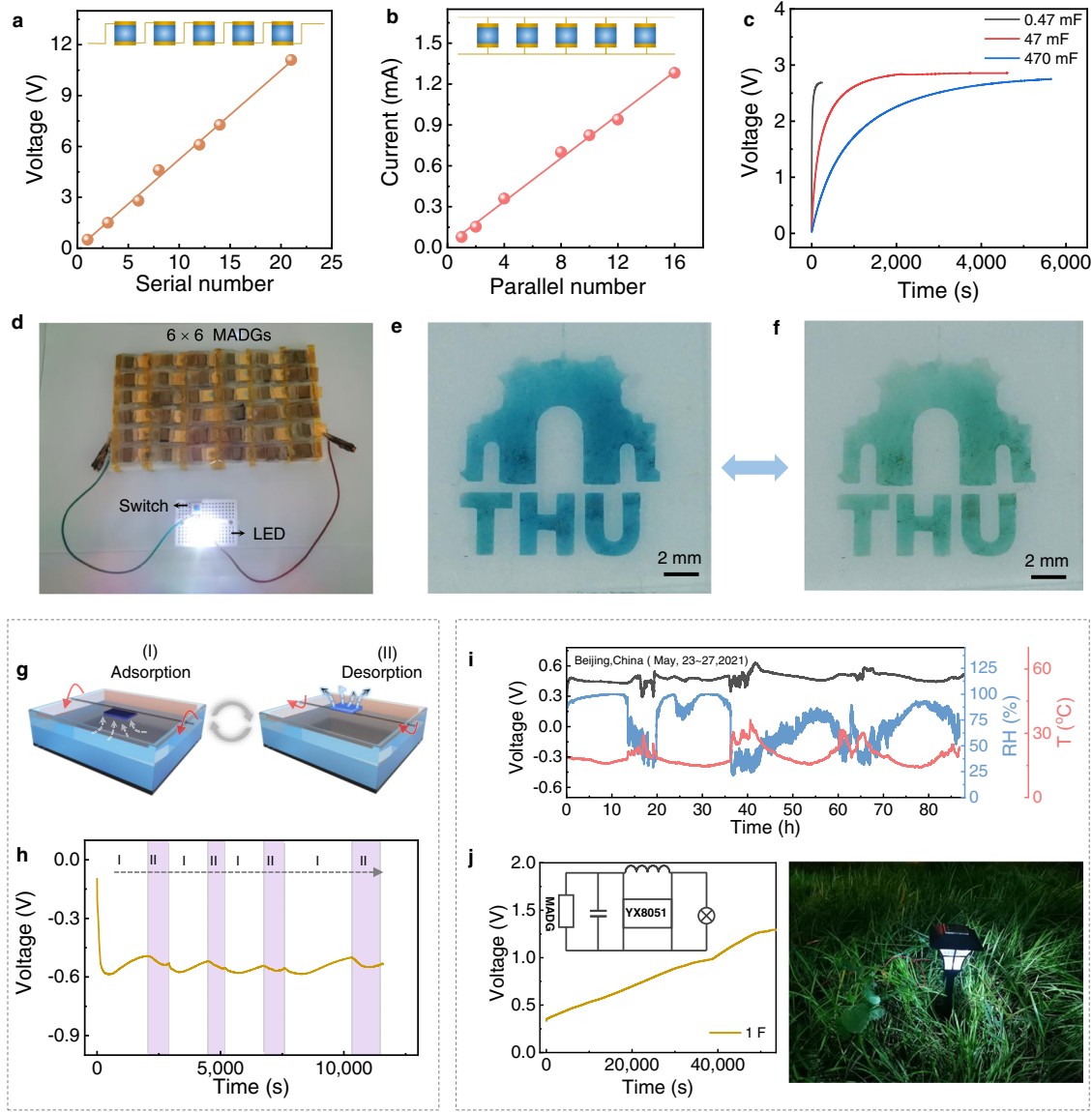

**Fig. 7 Applications of MADG. a** The relationship between open-circuit voltage and serial number of MADG units. The inset is the serial circuit. **b** Plot of short-circuit current with different number of MADG units in parallel. The inset is parallel circuit. **c** Voltage-time curves of commercial capacitors with various capacitance (0.47, 47 and 470 mF) charged by MADGs. **d** White LED powered by 6 × 6 integrated MADG device (90 ± 10% RH and 35 ± 5 °C). Digital photos of patterned PNAI film under **e** oxidized state and **f** reduced state. For the electrochromic process of PANI film, the MADG is directly served as electricity source. **g** Schematic diagram of designed auto-switchable adsorption and desorption generating setup. A water box is served as water source, and the MADG attached to a rotatable plate, could realize the dynamic switch of adsorbing and desorbing water molecules by rotation. **h** Constant voltage output of the designed set up. **i** Voltage variation generated by a MADG unit during outdoor testing (Beijing in China: East longitude ≈116° and North latitude ≈ 40°; Time on May 23th~27th, 2021). **j** Schematic of the circuit design and working demonstration of the self-powered road lamp. The generated electric energy is stored into 1 F capacitor in the outdoors (~25%−90% RH and 15°C−30°C).

## Methods

**Synthesis of GO.** GO solutions were synthesized by oxidation of natural graphite powder using a modified Hummers method[18]. Typically, 8 g graphite powder (325 mesh) and 4 g NaNO$_3$ were slowly added to 200 mL H$_2$SO$_4$ (98 wt%) and mechanically agitated for 0.5 h under the condition of ice water bath. 32 g KMnO$_4$ was slowly added to above mixed solution and agitated for 2 h. Then, 300 mL deionized water was slowly added and the mixture was agitated at 98ºC for 40 min. An additional 500 mL deionized water was added and 40 mL H$_2$O$_2$ (30 wt%) was added. Above mixture was filtered and washed with HCl (5 wt%). The obtained slurry was dispersed in 1,000 mL deionized water and subjected to dialysis (cut off of 8000~14,000 Da) for 15 days. Finally, the GO suspension was purified by centrifugation at 2000 rpm and 10,000 rpm.

**Preparation of SiO$_2$ nanofibers.** The SiO$_2$ nanofibers were fabricated through electrospinning technology[19]. Briefly, 20 g tetraethyl orthosilicate (TEOS), 0.14 g H$_3$PO4, and 20 g H$_2$O were uniformly mixed by magnetic stirring at room

temperature for 10 h. Aqueous polyvinyl alcohol (PVA, Mw ~15,000) dispersion (10 wt%) was obtained by dissolving 4 g PVA into 36 g water at 90ºC for 5 h under stirring. The mixed TEOS solution was added into PVA solution, and such mixtures was magnetic stirring for 5 h at room temperature. Then, the mixtures were electro-spun under a positively voltage of 18 kV and negatively voltage of −3.8 kV with a 22-gauge needle tip. The feeding rate was set as 0.8 mm min$^{-1}$. By further annealing treatment, the TEOS/PVA nanofibers could be changed into SiO$_2$ nanofibers at 850 °C for 2 h in air by gradually increasing the temperature at a heating rate of 10 °C min$^{-1}$. 2 g SiO$_2$ nanofibers were dispersed in 333.3 mL deionized water by homo-genizing the mixture for 30 min at 12000 rpm, which yielded uniform nanofiber dispersions (6 mg mL$^{-1}$).

**Preparation of porous SAG film.** 7 g aqueous SA solution (CP ~200 ± 20 mpa·s, 20 mg mL$^{-1}$), 6.7 g SiO$_2$ nanofibers (6 mg mL$^{-1}$), 2.2 g GO (9.1 mg mL$^{-1}$), 4.1 g water and 0.5 mL ethanol were uniformly dispersed by stirring. Then, the mixed solution was cast into petri dish (diameter of 55 mm). The porous aerogel was

prepared by freeze-drying of as-prepared mixtures. The GO component of aerogel was further reduced into rGO by the reduction of hydrazine hydrate vapour for 2 h at 80 ºC. Next, the reduced aerogel was immersed in aqueous AlCl₃ solution (0.3 M) to form ionic cross-linked aerogel. As-prepared cross-linked aerogel was dried in oven at 45ºC for 12 h, and was compressed under a pressure 10 MPa to obtain a flexible SAG film.

**Fabrication of MADG**. The SAG film was used as electricity-generating materials. The commercial gold foil was directly used as gold electrode without holes. And the gold electrode with holes is adopted gold-coated-stainless steel mesh by gold sputtering treatment. The gold electrode with/without holes was used as the upper/ bottom electrode of device, respectively. The SAG film was sandwiched between gold electrode, which was sealed by hot-melt resin except for the upper electrode, simply obataining MADG device.

**Electric measurements**. The electricity-generating performance measurements were performed with Keithley 2612B (Keithley Instruments, Cleveland, OH.). The circuit parameter of open-circuit voltage test was set current to 0 nA. The circuit parameter of short-circuit current test was voltage of 0 V.

**Self-powered electrochemical process**. For self-powered electrochemical polymerization of PANI, the MADG served as energy source, was connected with electrochemical cell, as well as platinum and FTO glass covered with mask layer is used as working and counter electrode, respectively. FTO glass was ultrasonically washed by acetone, ehanol and deionized water in sequence. After dried by nitrogen purge, the FTO glass covered with polyimide tape mask was immersed into an electrolyte solution. The electrolyte solution was composed of 0.5 M Na₂SO₄, 0.5 M H₂SO₄, and 0.05 M aniline. Before eletrochemical polymerization, the nitrogen was bubbled into electrolyte solution for 30 min. When the MADGs supply voltage and current between electrode pairs is about 1 V and 67 μA, PANI could be electro-polymerized on FTO. For self-powered electrochromic process, the PANI onto FTO electrode, platinum, and 1 M H₂SO₄ aqueous solution was used as working electrode, counter electrode, and electrolyte solution, respectively. When the MADG supply voltage (0 to 0.9 V) for electrochomic system, the oxidation process of PANI film was driven. When the MADG supply voltage (−0.8 to 0 V) for electrochomic system, the reduction process of PANI film was driven. During above oxidation/reduction processes, the colour of PANI film could change.

## Data availability
The data generated in this study are provided in the Source Data file and its Supplementary Information.

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

## Acknowledgements

This work was supported by the financial support from the National Science Foundation of China (No. 22035005, 52022051, 22075165, 52073159, 52090032), State Key Laboratory of Tribology (SKLT2021B03), Tsinghua-Foshan Innovation Special Fund (2020THFS0501). F. Liu also acknowledge the financial support from National Natural Science Foundation of China (11972349, 11790292), the Strategic Priority Research Program of the Chinese Academy of Sciences (XDB22040503). The authors are grateful for Senior Engineer, Haijun Yang and Dr. Yulan Tian for assisting solid state NMR tests at Tsinghua University, China.

## Author contributions

L.Q., H.C. and H.W. designed the experiments and accomplished the original draft. F.L. conducted the theoretical calculation. T.H., X.H., Y.H. and H.Y gave advice on experiments. All authors discussed the results and reviewed the manuscript. L.Q. supervised the entire project.

## Competing interests

The authors declare no competing interests.
