## [Peer Review File · Nature Communications]

Moisture adsorption-desorption full cycle power generationREVIEWER COMMENTS

Reviewer #1 (Remarks to the Author):

In this work, authors introduce a self-adaptive moisture adsorption-desorption power generator that can output high electricity. This device can spontaneously adsorb moisture at high RH and desorbs moisture at low RH, which fits the real application condition, indicating the potential for real usage. In general, the concept is impressive, and the performances are strong. However, several concerns need to be addressed before it can be published in Nature Communications. Please see the following detailed comments.

1. In the "Abstract" the authors explain that "A MADG unit could generate a high open-circuit voltage of ~ 0.5 V and a short-circuit current of $50\sim 100$ μ A whether at 100% RH or $15\pm 5\%$ RH". The reviewer is confused that whether 100 μ A is for $15\pm 5\%$ RH and 50 μ A is for 100% RH? However, the authors explain clearly in the "Introduction".
2. Only inspired by the phenomenon of intelligent adsorption and transpiration of Tillandsia? The basic working mechanism of the device in this work is also bio-inspired?
3. Why the open-circuit voltage under different RH is ~ 0.5 V, but the short-circuit current is different?
4. Please provide a picture for a "real device" in Figure 1. What are the specific dimensions?
5. In Figure 1d, is resin as the encapsulation layer? The location of resin is not clear to the reviewer.
6. The authors claim that "No deformation or breakages were observed after immersing a piece of the film into deionized water with stirring for 5 days (Fig. 2c)". Mechanical test results should be provided as a proof.
7. Why setting the experimental temperature of around 40 $^{\circ}$ C? To increase the desorbing process? So, what's the performance under different temperature?
8. What's the energy generating duration time when the device is fully saturated with moisture? Can the device maintain at peak output (like 100 μ A) when be fully saturated with moisture?
9. In Figure 3f, the reviewer thinks that comparing the power density with mW/cm^3 is more reasonable. The volume is more important than size.
10. For lighting LED demo in Figure 6 and the video, the reviewer suggests showing the whole system, including the device, the moisture supplier, and other things. This way can more clearly show the real condition during the experiment. "Clear" and "Real" are more important than "Beautiful".
11. What's the general and potential strategy to further improve the performances (like outputs especially voltage, stability, reliability, etc.) of such self-adaptive moisture adsorption-desorption moisture generator? Please explain something in the "Discussion", even though the "Optimization of performance" is done.
12. One suggestion, the reviewers thinks that the demo of "MADG-driven electrochemical polymerization or electrochromic of PANI" is somehow not so important. This demo seems has no practicability. I appreciate the authors put a lot of efforts in explaining the mechanism, this is important and fundamental. In Figure 6, demonstration of a self-powered road lamp is enough and closer to the practicability.

Reviewer #2 (Remarks to the Author):

The paper (NCOMMS-21-37604) proposes an interesting moisture generator for collecting energy in wet environment with sustainable power output. Compared to previous devices that can only give transient currents, the authors have done an excellent job in achieving continuous power output, which is significant and groundbreaking. However, the paper is completely overloaded with information. That makes it hard to read; at least if you want to understand also the details which are mentioned. I propose that the figures are reduced to make it more concise. I appreciate that the authors want to offer every detail of the study, but again it should be formulated more concise. I am still confused about the mechanism part which is the major weakness in the current form. Overall, I think the paper is very interesting and an important contribution and it can be published after reasonable major revisions.

Some comments are listed:

- 1) There are many hygroscopic systems in both natural and anthropic environments. Thus, the title seems a little bit pretentious and too much appealing. On the other hand, perhaps authors can explain why this system is unique and "bioinspired".
- 2) As mentioned, the authors have done an excellent job in achieving significant power output. In fact, authors claimed that: "...establishing an unprecedented trade-off between internal resistance and output power density". This is great, but the material fabrication seems really complicated and time-consuming. How is this device compared in terms of cost-benefit with recent reported devices such as [Energy Environ. Sci., 2018, 11, 2839–2845] or [Energy Environ. Sci., 2021,14, 353-3580] that have roughly similar performance?
- 3) The present article provides experiments and theoretical models trying to explain moisture-adsorbing power generation. On the other hand, this Reviewer has two big concerns:
 - a. The working principle was speculated by molecular dynamics (MD) simulations and Monte Carlo calculation. How is the ions/water molecules ratio? The impression is that there are many ions per water molecule so that this is far from representing experimental data.
 - b. Authors have provided many experiments to verify the potential on SAG film (zeta, KPFM and etc). On the other hand, authors have missed the opportunity of using KPFM under different humidities. Scanning SAG film under low humidities and thus at some point increase the water content can provide valuable information on moisture-adsorbing electricity build-up.
- 4) Why is the temperature so high (40°C) in many experiments? Is it an optimized testing condition? If so, this would limit the applicability of this device.
- 5) Although the applications of MADG are very impressive, comparisons with other devices are always difficult to achieve. On the other hand, many moisture-based power generators have been reporting the voltage and electric current on different resistors connected to the circuit as load. This seems to adequate to this work. Also, authors could report the electric current of MADG through a load resistor uninterruptedly, maybe for 90 hours as is presented in Fig. 6(i).

Reviewer #3 (Remarks to the Author):

The manuscript by Haiyan Wang et al. proposed a moisture adsorption-desorption power generator (MADG), which demonstrates a cyclic electric output in a dynamic humidity by adsorbing moisture at high RH and desorbing moisture at low RH. Generally, making use of the humidity difference is an interesting approach to address the unsustainable and non-repetitive issues of moisture generators. However, it is not suitable to be claimed as self-maintained, which actually relies heavily on switchable humidity or large humidity difference of environment. Another point of this manuscript is, different from their former power-generating mechanism, the authors proposed a new principle based on ions diffusion driven by ions concentration difference and ion-hydration energy. However, there should be

only one major mechanism responsible for these materials with carboxyl functional groups. As to the proposed mechanism, there are still several issues unrevealed especially related to the ion-hydration process. At this current stage, it is not convincing to explain the absorption and desorption process in different humidity. The following issues related to the proposed process should be fully addressed, and direct evidence should be provided.

(1) The calculation results in Fig. 2e and f are not reasonable. It seems to me that these data were calculated based on the tested curves of Fig.2d, however, most data in different RH states have not reach the equilibrium state, therefore the obtained Me is not reasonable to be used for the calculation as described in Equation (1) and (2). Also, the authors should explain why the SAG film displays the highest adsorption diffusion coefficient (D) in the low RH of 20% (Fig. 2f). Generally, the SA of porous SAG network that plays main role of water absorption demonstrates much higher water absorption capacity and dynamic in high RH than in low RH at 40oC.

(2) Based on the theoretical calculation, the authors claimed the ion-hydration energy of ion-water cluster serves as a driving force, which allows the directional ions movement from the moisture-releasing region to well-watered region. As to the proposed ions concentration difference and ion-hydration energy, which driving force is dominant during absorption and desorption process, respectively, and why?

(3) As described in the manuscript, the MADG yields spontaneous ions flow in the same direction during adsorption and desorption process. The induced voltage and current also display the same direction in the high and low RH condition. In this case, does it mean Na⁺ ions would finally run out from one side and accumulate at another side? What would happen in the end after cycles? In this case, it is not suitable to be claimed as closed-loop pathway.

(4) We also noticed that the authors' earlier work on a bilayer film that composed of an analogous material (PSSA) as electricity generator. Actually, as the main part responsible for the electricity generation, PSSA has similar carboxyl functional group with SA in this manuscript, which should share the same electricity-generating principle. However, the authors proposed a different mechanism in these two reports. The authors should clarify it.

(5) As described in manuscript, the authors claimed the force of ion-hydration energy might surpass that of ions concentration difference and presumably dominate ions migration in low RH condition. While in high RH condition, it was mainly dominated by ions concentration difference. Either Fig.1d or Fig. 3a illustrate the ion-hydration process in the high RH. Therefore, it seems the proposed mechanism is paradox when explain the absorption and desorption process. The author should provide solid evidence.

(6) Actually, there are two kinds of Na⁺ ions in the system. One is the dissociated free Na⁺ ions, and another is Na ions that still anchored on the polymer chains. The author should make it clear which kinds of the Na⁺ ions cline to be hydrated and why? It seems this supposed mechanism only take the dissociated free Na⁺ ions into consideration.

(7) Does this system rely heavy on a high humidity difference (ΔRH)? The author mainly demonstrates the feasibility at extreme condition by changing the RH from 90 \pm 10% to 15 \pm 5% at 40 oC, that is, induced by a high RH difference ($\Delta RH=80\%$). However, its universality in other RH condition needs to be verified. For example, whether lower initial RH states (i.e., change from RH= 35% to 15%) or smaller RH difference (i.e., changing from RH=100% to 80%) still work and obtain the same conclusion? Besides, is there any critical RH difference that is required before to achieve sustained performance?

(8) Why the temperature of 40 oC was chosen to characterize the moisture adsorption and desorption kinetics and electricity generation performance of SAG film? It has been known that the diffusion dynamic largely related to temperature, how the electricity generation performance at RT? According to Supplementary Fig. 1, the variation of relative humidity (RH) and temperature is large, the author should take the temperature' effect into consideration.

(9) The authors claimed a universality of the moist-adsorb-desorb-generation; however, Fig.4 f only presents the voltage output for several analogous materials, how about their short-current values?

(10) Fig. 3c and d. Is there any method or strategy to control the initial open voltage always start exactly from 0 V?

(11) According to Fig.4a, the described hierarchical pore structure of SAG film seems play important

role to facilitate water molecules and ions transport during the absorption and desorption process, therefore the corresponding BET related to the porosity of SAG and SA-rGO should be provided.

(12) The performances in Fig. 6l were not consistent with the results obtained in Fig. 3c, d, g and h. For example, Fig. 3c delivers a voltage output (0.15V) after lasted for 3.3 h when maintained in high RH condition (100%), however, Fig. 6l shows the voltage output maintains at $\sim 0.45V$ after 14.5h during the first period in a relative stable RH condition close to 100%. The author should explain it.

(13) The authors should explain the fluctuation of voltage in Fig. 3g. Besides, why the voltage in Fig. 3h is relative stable while th

Point-by-Point Responses to the Reviewers' Comments

Reviewer #1 (Remarks to the Author):

In this work, authors introduce a self-adaptive moisture adsorption-desorption power generator that can output high electricity. This device can spontaneously adsorb moisture at high RH and desorb moisture at low RH, which fits the real application condition, indicating the potential for real usage. In general, the concept is impressive, and the performances are strong. However, several concerns need to be addressed before it can be published in Nature Communications. Please see the following detailed comments.

Reply: We thank the referee for the positive comments and constructive suggestions for improving our manuscript. We have conducted additional experiments and added more discussions in the revised manuscript. Some modifications and explanations are summarized as below.

1. In the “Abstract” the authors explain that “A MADG unit could generate a high open-circuit voltage of ~0.5 V and a short-circuit current of 50~100 μ A whether at 100% RH or 15 \pm 5% RH”. The reviewer is confused that whether 100 μ A is for 15 \pm 5% RH and 50 μ A is for 100% RH? However, the authors explain clearly in the “Introduction”.

Reply: The generated current is ~100 μ A at high RH (100%) environment and ~50 μ A at low RH (15 \pm 5%) condition. Per your suggestion, we have modified the sentence as “A MADG unit can generate a high voltage of ~0.5 V and current of ~100 μ A at 100% RH, and deliver electric output (~0.5 V and ~50 μ A) at 15 \pm 5% RH” in the revised manuscript (Page 2) for clear understanding.

2. Only inspired by the phenomenon of intelligent adsorption and transpiration of Tillandsia? The basic working mechanism of the device in this work is also bio-inspired?

Reply: Thanks for the insightful comments. The moisture adsorption and transpiration of Tillandsia and resultant directional charges diffusion in the cells of Tillandsia are both inspiration sources for MADG device design. First, Tillandsia species adaptively conduct water adsorption or transpiration in air in response to dynamic RH. Inspired by this phenomenon, we design a self-adaptive moisture adsorption-desorption power generator, which enables to desirably match dynamic RH environment. Second, in the cells of Tillandsia, directional water transport induced by adsorption at high RH or desorption at low RH, will induce directional ions diffusion, leading to the generation of electricity in organism (Figure R1, *Nature*, 2017, 552, 214–218; *Nat. Nanotechnol.*, 2008, 3, 666–670). Inspired by this mechanism, the proposed device for electricity generation adopts ionizable and porous assembly served as electricity-generating materials with asymmetric package, which allows directionally ions diffusion and induced electric output in response to moisture adsorption and desorption. In general,

the designed MADG is based on the inspiration of both adsorption-desorption phenomenon and directional charges diffusion mechanism.

As suggested, above bio-inspired design principle of the MADG device has been added in the revised manuscript (Page 2–4 and Supplementary Information Page 8) and highlighted.

Figure R1. Schematic of self-adaptive water adsorption or transpiration of Tillandsia and working mechanism of bioelectric generation in Tillandsia cell.

3. Why the open-circuit voltage under different RH is ~0.5 V, but the short-circuit current is different?

Reply: In fact, the electric output of MADG device could be generated by both moisture adsorption and desorption process, and the open-circuit voltage and short-circuit current vary with RH condition. The initially dry MADG device can spontaneously adsorb moisture and accordingly induce electric signals at different RH environment. As shown in Figure R2a–c, both voltage and current outputs are enhanced as the RH increases in the moisture adsorption power generation, because high RH condition is beneficial to water adsorption and improve the uptake amount in device. The MADG device saturated with moisture could further conduct moisture desorption process for power generation. As shown in Figure R2d–f, the generated voltage and current outputs raises as the RH decrease in the moisture desorption power generation, arisen from that low RH environment enables to facilitate water desorption in device.

Related discussion of the RH influence on performance has been added in revised manuscript (Page 13) and Supplementary Information (Page 24).

Figure R2. (a, b) Voltage and current output of device versus environmental RH in the moisture adsorption power generation (40°C). (c) The mass change of SAG film by water molecule adsorption upon varied RH under 40°C. (d, e) Voltage and current output of saturated device at different RH condition for moisture desorption power generation (40°C). (f) The mass change of saturated SAG film by water molecule adsorption upon varied RH under 40°C.

4. Please provide a picture for a “real device” in Figure 1. What are the specific dimensions?

Reply: According to your suggestion, we have added “real device” Figure R3 to Supplementary Fig. 17 (Supplementary Information Page 15) and related analysis to revised manuscript (Page 10).

Figure R3. Photograph of a MADG device, which is composed of electricity-generating layer with size of $1.00 \times 0.89 \times 0.02 \text{ cm}^3$ sandwiched between a pair of gold electrode, and sealed with resin.

5. In Figure 1d, is resin as the encapsulation layer? The location of resin is not clear to the reviewer.

Reply: The resin is used as the encapsulation layer in Fig. 1d. As suggested, we have improved the diagram to make the location of resin clear in Fig. 1d in the revised manuscript (as shown in Figure R4).

Figure R4. Scheme of the device structure of MADG device. The MADG device is composed of gold electrodes, ionizable porous assembly as electricity-generating material, and encapsulation layer.

6. The authors claim that “No deformation or breakages were observed after immersing a piece of the film into deionized water with stirring for 5 days (Fig. 2c)”. Mechanical test results should be provided as a proof.

Reply: As suggested, the mechanically compressive test of SAG in water condition is displayed in Figure R5. The compressive stress versus strain curves of SAG have no visible changes before and after 5 days immersing into water, demonstrating the mechanical stability. Further analysis (Figure R6) indicates that the morphologies and pore structure are lack of obvious change before and after 5 days in water. These results consistently illustrate the SAG film is highly stable in the water.

Figure R5 and R6 have been added to Supplementary Fig. 8 (Supplementary Information Page 11) and corresponding analysis and discussion in the main text (Page 6).

Figure R5. Compressive stress versus compressive strain for SAG before and after 5 days immersing into water. Insets are the photos of SAG in water before and after 5 days.

Figure R6. Scanning electron microscopic (SEM) images at different magnification for SAG film before (a–c) and 5 days (d–f) immersing in water.

7. Why setting the experimental temperature of around 40 °C? To increase the desorbing process? So, what's the performance under different temperature?

Reply: Thanks for your valuable comments. The experimental temperature of around 40°C is a proper control condition we have tried. Considering the common temperature range in nature, we have investigated the power-generating performance of MADG under different temperature from 25°C to 40°C. As shown in Figure R7, when the value of temperature from 25°C to 40°C (~100% RH), the generated voltage of MADG increases from 0.35 to 0.53 V, and current output rises from 27 to 82 μA. It is attributed that relatively high temperature is beneficial to water adsorption and improve the rate of adsorbing water molecules at 100% RH (Figure R8a). Meanwhile, the induced voltage and current output by moisture desorption process are also enhanced with the temperature from 25°C to 40°C (15% RH, Figure R7d), because proper temperature enables to rise the rate of desorbing water molecules at 15% RH (Figure R8b), leading to increasing the desorbing process. In addition, the electrochemical impedance spectroscopy displays that the ionic resistance visibly decreases with increasing temperature (Figure R8c), replying enhanced ions transport under relatively high temperature. Above results also suggest that MADG device is able to exert electricity-generating performance under commonly ambient temperature (25°C~40°C). By the way, since the environmental temperature would be usually below 40°C, the electric output of MADG is not further investigated at higher temperature condition.

In essence, the influence of temperature on electricity generation is arisen from affecting water molecules and free ions transport. The MADG device is able to exert adsorption and desorption enabled power generation under different temperature. We set the temperature of around 40°C, which is just served as a proper experimental control condition to systematically investigate the process of moisture adsorption-desorption generation.

Related discussion of the temperature influence on performance has been added in revised manuscript (Page 13) and Supplementary Information (Page 25).

Figure R7. The voltage-time (a) and current-time profile (b) in moisture adsorption electricity generation under different temperature ($\sim 100\%$ RH). The generated voltage and current in moisture adsorption power generation (c) and desorption power generation (d) under different temperature.

Figure R8. (a) Rate of adsorbing water molecules during moisture adsorption under different temperature and $\sim 100\%$ RH. (b) Rate of desorbing water molecules during moisture desorption under different temperature and $\sim 15\%$ RH. (c) The electrochemical impedance spectra of MADG device under different temperature and $\sim 100\%$ RH.

8. What's the energy generating duration time when the device is fully saturated with moisture? Can the device maintain at peak output (like $100 \mu\text{A}$) when be fully saturated with moisture?

Reply: In this study, the MADG device enables to adsorb moisture and induce the formation of electric signal for tens of thousands of seconds under 100% high RH environment (Figure R9a). At final state of moisture adsorption, the MADG device will fully saturate with moisture and approach to adsorption equilibrium, leading to balanced ions diffusion and zero-output. As a result, the device is unable to permanently maintain at peak output (like $100 \mu\text{A}$) at high RH. But the MADG device saturated with moisture could exert moisture desorption power generation under 15% low RH, and its duration time is about 9000 s for single desorption process (Figure R9b).

Above results could be explained by working principle of MADG. The working process of MADG device consists of two parts: moisture adsorption enabled power generation at high RH and moisture desorption induced power generation at low RH. In high RH environment, the device can asymmetrically adsorb moisture and dissociate free ions, which enable to diffuse from high concentration to low concentration driven by ions concentration difference, thus giving rise to electric signal for long time on the external circuit (Figure R9a). The electric output associates well with the water adsorption process. With adsorbed moisture saturation of device, the ions diffusion will be balanced and MADG device will reach into the equilibrium state and stop electricity output. Subsequently, the MADG device saturated with moisture could conduct moisture desorption power generation under low RH condition. In response to asymmetric desorption of water molecules, a net ions flow could be formed driven by ion-hydration energy in MADG, inducing electric signal, which also relies on the rate of moisture desorption (Figure R9b). The moisture desorption power generation will approach to balanced state and stop along with desorption equilibrium. Based on the working principle of MADG, the moisture adsorption power generation is associated with moisture-adsorbing rate, thus above process could be accelerated by directly dropping water into device. Figure R9c and d display the electric output of saturated device induced by dropping water at ambient condition. Such saturated device could maintain electric output for about 18,000 s in single power generation cycle.

Related explanation of working processes of device has been added in the revised manuscript and highlighted (Page10).

Figure R9. (a) Change in mass ($\Delta m = m/m_0 \times 100\%$, m and m_0 denotes the mass of adsorbed or desorbed water and initial mass of sample, respectively), adsorption rate ($d(\Delta m)/dt$), current and voltage during moisture adsorption power generation under $\sim 100\%$ RH and 40°C . (b) Change in mass (Δm), desorption rate ($-d(\Delta m)/dt$), current and voltage during moisture desorption power generation under $\sim 15 \pm 5\%$ RH and 40°C . (c) Schematic illustration of electricity-generating device by dropping water. (d) Voltage output of MADG device by directly dropping water.

9. In Figure 3f, the reviewer thinks that comparing the power density with mW/cm^3 is more reasonable. The volume is more important than size.

Reply: Thanks for your insight points. Per your suggestion, the output volumetric power density of MADG device in this work has been compared with that of reported device, as summarized in Table R1. Figure R10 indicates that the MADG device has a good trade-off between internal resistance and output volumetric power density, prior to that of the reported device. It should be pointed that there are lack of descriptions of specific size of device in many reported papers. And most of the reported papers only represent the area or thickness of electricity-generating materials, and no information on size of electrodes. Thus, the mentioned output volumetric power density is calculated by the volume of electricity-generating material.

The analysis of output volumetric power density has been added in the main text (Page 11 and 12), related data has been added as Supplementary Table 4 and Fig. 23 (Supplementary Information Page 19 and 20).

Table R1. Summary of output volumetric power density of water-related power generators.

NO.	Material	Volume (cm ³)	R (kΩ)	P _{output} (μW cm ⁻³)	Reference
1	Biological nano-fibrous	0.2	10,000	0.003	Adv. Funct. Mater. 2019, 29, 1901798
2	Ni-Al layered double hydroxide	0.003	2,000,000	15	Nano Energy 2020, 70, 104502
3	Fabric coated with carbon black	0.32	63,000	0.78	ACS Nano 2019, 13, 12703–12709
4	Ni-Al layered double hydroxide	0.011	270,000	16.1	Nano Energy 2019, 57, 269–278
5	Printable carbon film	0.021	1,500,000	8.1	Adv. Funct. Mater. 2017, 27, 1700551
6	Graphene oxide membrane	4.8 × 10 ⁻⁴	10,000,000	1.5	Energy Environ. Sci. 2018, 11, 2839–2845
7	MoS ₂ film	5 × 10 ⁻⁴	3,000	42	Nano Energy 2021, 81, 105630
8	Bilayer of polyelectrolyte film	0.005	20,000,000	0.9	Nat. Nanotechnol. 2021, 16, 811–819
9	Graphene oxide and sodium polyacrylate	0.001	10,000,000	7	Energy Environ. Sci. 2019, 12, 1848–1856
10	Carbon-coated cotton fabric	0.32	20,000	6.2	Energy Environ. Sci. 2020, 13, 527–534
11	Gradient graphene oxide and graphene oxide	0.0016	10,000,000	11.1	Nat. Commun. 2018, 9, 4166
12	Corn stalk	0.6	741,100	0.89	Nano Energy 2020, 74, 104922
13	Protein nanowires	1.75 × 10 ⁻⁴	1,000,000	266	Nature 2020, 578, 550–554
14	Poly(4-styrenesulfonic acid) membrane	0.02	24,400	850	Energy Environ. Sci. 2019, 12, 972–978
15	SA-SiO ₂ -rGO	0.018	4,000	600	This work

Figure R10. Comparison of output volumetric power density and internal resistance between MADG device in this work and reported water induced generators. The numbers correspond to reference numbers in Table R1.

10. For lighting LED demo in Figure 6 and the video, the reviewer suggests showing the whole system, including the device, the moisture supplier, and other things. This way can more clearly show the real condition during the experiment. “Clear” and “Real” are more important than “Beautiful”.

Reply: According to your valuable suggestion, we have supplemented the whole system of lighting LED demo in Figure 6 (Figure R11) and Supplementary Video 1. Figure R11 has been to Fig. 7d (Page 20) and Supplementary Fig. 44 (Supplementary Information Page 29), and corresponding discussion has been added in the main text (Page 19 and 20).

Figure R11. Six red LEDs (a) and white LED (b) powered by 6×6 integrated device.

11. What’s the general and potential strategy to further improve the performances (like outputs especially voltage, stability, reliability, etc.) of such self-adaptive moisture adsorption-desorption moisture generator? Please explain something in the “Discussion”, even though the “Optimization of performance” is done.

Reply: Thanks for your good suggestions. In essence, the MADG device reasonably enables to yield directional ions flow driven by ions concentration difference and ion-hydration energy, triggered by asymmetric moisture adsorption or desorption, respectively, thus producing free electron movement of the external circuit. Therefore, the key points for improving performance are how to achieve rapid moisture adsorption and desorption, abundant ions dissociation, effective positive-negative charge separation, and fast ion diffusion. And the general and potential strategies could be divided into four aspects: 1) Electricity-generating material regulation; 2) Device structure design; 3) Development of scalable integration technology.

1) Electricity-generating material regulation

The regulation strategy involves from micro to macro scale to develop desirable power-generating materials. First, on the molecular scale, improving the density of functional group and constructing aligned molecular chains enable to facilitate ions dissociation and conduction (*Nat. Mater.*, 2018, 17, 725–731). Second, on the micro-structure, hierarchically porous and robust structure, constructed by single or hybrid nanomaterials ranging from 0D, 1D, 2D, to bulk, can reduce lamellar stacking and promote the exposure of functional groups to form efficient moisture adsorption and desorption (*Adv. Mater.*, 2020, 32, 2003722). And the vertically orientated channel is capable for shortening the pathway of water molecules and ions transport, and advocating mass transfer (*Nature*, 2018, 557, 409–412). Third, a range of macroscopic properties need to be tuned, including water-solubility,

hydrophilicity, conductivity, mechanical and chemical stability, need to be regulated, which also have significant influence on power generation.

2) Device structure design

The device structure could be optimized through the following aspects, including encapsulation layer, one-all-in structure, and circuit design. Optimal encapsulation layer will advocate asymmetrical moisture adsorption and desorption. One-all-in structure renders a stable contact between electrode and material, which is beneficial to stability and reliability of device. Reasonable circuit design makes it possible for signal rectification and stable output.

3) Development of scalable integration technology

Apart from unit optimization, the development of scalable integration technology is also essential to scale up electric output and practical application. Advanced manufacturing technology will be useful for large integration, such as laser processing, 3D printing, and spraying.

In the revised manuscript, as suggested, we have added related outlook on the strategy of improving performance in the “Discussion” part (Page 23 and 23).

12. One suggestion, the reviewers think that the demo of “MADG-driven electrochemical polymerization or electrochromic of PANI” is somehow not so important. This demo seems has no practicability. I appreciate the authors put a lot of efforts in explaining the mechanism, this is important and fundamental. In Figure 6, demonstration of a self-powered road lamp is enough and closer to the practicability.

Reply: We agree with this point. We have modified and simplified Fig. 6 and related discussions in the revised manuscript (Fig. 7, Page 19–21).

Reviewer #2 (Remarks to the Author):

The paper (NCOMMS-21-37604) proposes an interesting moisture generator for collecting energy in wet environment with sustainable power output. Compared to previous devices that can only give transient currents, the authors have done an excellent job in achieving continuous power output, which is significant and groundbreaking. However, the paper is completely overloaded with information. That makes it hard to read; at least if you want to understand also the details which are mentioned. I propose that the figures are reduced to make it more concise. I appreciate that the authors want to offer every detail of the study, but again it should be formulated more concise. I am still confused about the mechanism part which is the major weakness in the current form. Overall, I think the paper is very interesting and an important contribution and it can be published after reasonable major revisions.

Reply: We greatly thank the referee for positive recommendation and constructive suggestions for correcting and improving our manuscript. We have conducted additional experiments and theoretical calculations, and supplemented more discussions in the revised manuscript, especially for working mechanism of MADG. We also have carefully simplified and refined figures and main text. Some modifications and explanations are summarized as below.

Some comments are listed:

1) There are many hygroscopic systems in both natural and anthropic environments. Thus, the title seems a little bit pretentious and too much appealing. On the other hand, perhaps authors can explain why this system is unique and “bioinspired”.

Reply: Thanks for your valuable comments. The moisture adsorption and desorption is indeed common phenomenon in both natural and anthropic conditions. Inspired by this phenomenon, a self-adaptive moisture adsorption-desorption power generator is proposed, which enables to desirably match dynamic RH environment. In fact, the variation of RH in a day is general natural phenomenon along with hydrological cycle. However, the previously reported moist-electric generator heavily depends on high RH, which makes it challenging for application in varied RH environment. Therefore, benefiting from integrating adsorption and desorption power generation into a closed-loop process, the designed all-new device could afford self-adaptive electricity-generating performance according to dynamic RH for the first time, indicating its novelty.

Furthermore, the working mechanism of designed device is also bio-inspired. As shown in Figure R1, directional water transport induced by adsorption at high RH or desorption at low RH, will induce ions diffusion, leading to the generation of electricity in the cells of *Tillandsia* (*Nature*, 2017, 552, 214–218; *Nat. Nanotechnol.*, 2008, 3, 666–670). Inspired by this mechanism, the MADG device for electricity generation harnesses ionizable and porous assembly served as electricity-generating materials with asymmetric package, which allows directionally ions diffusion and induced electric output in response to moisture adsorption and desorption.

In short, the bioinspired MADG device is based on the inspiration of both adsorption-desorption phenomenon and directional charge diffusion mechanism. Compared with previous MEG by utilizing a single adsorption process, such unique MADG device firstly integrates adsorption and desorption enabled power generation into a closed-loop process, thus affording self-adaptive and repeatable electricity-generating performance under dynamic RH condition.

As suggested, related discussion has been added in revised manuscript (Page 2–4) and Supplementary Information (Page 8).

Figure R1. Schematic of self-adaptive water adsorption or transpiration of Tillandsia and working mechanism of bioelectric generation in Tillandsia cell.

2) As mentioned, the authors have done an excellent job in achieving significant power output. In fact, authors claimed that: “...establishing an unprecedented trade-off between internal resistance and output power density”. This is great, but the material fabrication seems really complicated and time-consuming. How is this device compared in terms of cost-benefit with recent reported devices such as [Energy Environ. Sci., 2018, 11, 2839–2845] or [Energy Environ. Sci., 2021, 14, 353–3580] that have roughly similar performance?

Reply: Thanks for the comments. The electricity-generating film in current work consists of natural product of sodium alginate (SA), silicon dioxide nanofiber (SiO₂), and reduced graphene oxide (rGO), which are low-cost, general and harmless materials. By using freeze-drying and chemical reduction method, the hybrid film could be easily fabricated. Such fabrications are normal and facile methods, which are widely utilized in sorts of scalable production in experiments. In the regard, the used materials and fabrication methods in current work would be controllable. There seems to be lack of systematic evaluation of cost-benefit in other reported works. Certainly, we will focus on the cost-benefit of device in future research work.

More importantly, the obtained hybrid material performs remarkable superiorities compared to previous materials. Previous electricity-generating material, such as polyelectrolyte and graphene oxide-based material (*ACS Materials Lett.*, 2021, 3, 193–209; *Energy Environ. Sci.*, 2018, 11, 2839–2845), is almost lack of superbly stable micro-structure, resulting in dissolution itself in water and the

destruction of porous structure, which seriously degrades their cyclic performance and reliability. Besides, the regulation of internal resistance of electricity-generating material has been ignored all along, but it is a key parameter for direct power supply. SA featured with abundant carboxyl functional group, not only enables to construct a crosslinked polymeric network, but also is ideally utilized as electricity-generating material. Such SA chains uniquely combine with one-dimensional SiO₂ nanofiber and rGO nanosheets to assemble into robustly interconnected three-dimensional and ionizable skeleton with hierarchical microstructure and tunable resistance. Based on synergistic effect, this ionizable assembly can exert unprecedented water molecules transfer, ions dissociation, ions transport, and low internal resistance. Correspondingly, the hybrid film can render a considerable electric output and a good trade-off between internal resistance and maximum output power density, prior to that of the reported electricity-generating materials.

Peer your suggestions, we have added the related concerns and discussions in revised manuscript (Page 6) and highlighted.

3) The present article provides experiments and theoretical models trying to explain moisture-adsorbing power generation. On the other hand, this Reviewer has two big concerns:

a. The working principle was speculated by molecular dynamics (MD) simulations and Monte Carlo calculation. How is the ions/water molecules ratio? The impression is that there are many ions per water molecule so that this is far from representing experimental data.

Reply: The ion/water molecules ratio of SAG film has been investigated by solid state nuclear magnetic resonance (NMR) testing. We measured ²³Na solid state NMR of a series of aqueous sodium sulfate solutions with different concentrations (Figure R2) to obtain calibration curve of Na⁺ ion concentration (Figure R3a). By calibration curve method, the measured free Na⁺ ion/water molecules ratio is about 5×10^{-5} in hydrated SAG film with water uptake of 167% (Figure R3b), suggesting many water molecules around per Na⁺ ion. Notably, the water molecules number is far more than that of Na⁺ ions, but there are only dozens of water molecules could closely interact with dissociated Na⁺ ions according to ion hydration principle (https://en.wikipedia.org/wiki/Solvation_shell; *Nature*, 2018, 557, 701–705; *Adv. Colloid Interface Sci.*, 2019, 268, 1–24). In addition, if all Na atoms composed of both dissociated ions and bonded atoms anchored on polymer chains, are taken into consideration for the ratio calculation, there might be much more Na atoms compared to water molecules.

More importantly, only dissociated hydrated Na⁺ ions, combining with enough water molecules and overcoming the binding energy between Na⁺ and COO⁻, could diffuse freely in the system and thus induce electric output, while it is not the case for bonded Na atoms. As the major concern in this work is power generation, only dissociated Na⁺ ions are considered in our molecular dynamics (MD) simulations and kinetic Monte Carlo calculation. Physical properties of ion in water are determined by their corresponding hydrated ion, and it is this reason plenty of experimental and theoretical studies are devoted to obtain hydration energy (*J. Chem. Phys.*, 1980, 72, 260–263; *J. Phys. Chem.*, 1996, 100, 1206–1215; *J. Chem. Phys.*, 1997, 106, 9769–9780; *J. Braz. Chem. Soc.*, 2008, 19, 101–110; *J. Chem.*

Phys., 2003, 118, 7062–7073). Especially, a cylinder pore model is used to study ion hydration energy and thus deduce the nano-pore selectivity (*J. Chem. Phys.*, 2004, 93, 168104). Compared with this pore model, our calculation model has exactly the same physical consideration, only differs by the specific geometrical configuration around hydrated ions. Therefore, our calculation model should be reasonable.

Related discussion has been added in revised manuscript (Supplementary Information Page 3, 4 and 14) and highlighted.

Figure R2. ^{23}Na solid state NMR spectra of Na_2SO_4 solution with various concentration.

Figure R3. (a) Plot of integration of characteristic peak in ^{23}Na NMR with different concentration of Na_2SO_4 solutions. By linear fitting, corresponding calibration curve was obtained. (b) ^{23}Na solid state NMR spectra of 0.01 M Na_2SO_4 solution, hydrated SAG film, dry SAG film, and 0 M Na_2SO_4 solution. The ^{23}Na chemical shift of SAG film with water content of 167% is ~ 0 ppm, which accords with that of Na^+ ion of aqueous Na_2SO_4 solution.

b. Authors have provided many experiments to verify the potential on SAG film (zeta, KPFM and etc). On the other hand, authors have missed the opportunity of using KPFM under different humidities. Scanning SAG film under low humidities and thus at some point increase the water content can provide valuable information on moisture-adsorbing electricity build-up.

Reply: We are grateful for this comment. According to your suggestion, we have conducted kelvin probe force microscope (KPFM) tests on SAG film under different RH. As shown in Figure R4, the relative surface potential of SAG film is approximately 450 mV at 55% RH, and obviously rises to 610 mV at 100% RH. The surface potential of SAG film increases with the enhance RH, which implies more water capture and sodium ions dissociation (*Nat. Nanotechnol.*, 2021, 16, 811–819; *IEEE T. Dielect. El. In.*, 2017, 24, 1913–1922) in SAG film at high RH. Therefore, the results indicate that the ions dissociation and diffusion triggered by moisture adsorption in SAG film could be key processes for moisture-adsorbing electricity generation.

In the revised manuscript, we have supplemented related analysis as suggested (Supplementary Information Page 13).

Figure R4. (a, b) KPFM images (a) and relative surface potential (b) along the marked black line in 55% RH condition. (c, d) KPFM images (c) and relative surface potential (d) along the marked black line in 100% RH condition.

4) Why is the temperature so high (40°C) in many experiments? Is it an optimized testing condition? If so, this would limit the applicability of this device.

Reply: Thanks for your good points. The experimental temperature of around 40°C is a proper control condition we have tried. According to the common temperature range in nature, we have investigated the power-generating performance of MADG under different temperature from 25°C to 40°C. As

shown in Figure R5, when the value of temperature from 25°C to 40°C (100% RH), the generated voltage of MADG increases from 0.35 to 0.53 V, and current output rises from about 27 to 82 μA . It is attributed that relatively high temperature is beneficial to water adsorption and improve the rate of adsorbing water molecules at 100% RH (Figure R6a). Meanwhile, the induced voltage and current output by moisture desorption process are also enhanced with the temperature from 25°C to 40°C (15% RH, Figure R5d), because proper temperature enables to rise the rate of desorbing water molecules at 15% RH (Figure R6b), leading to increasing the desorbing process. In addition, the electrochemical impedance spectroscopy displays that the ionic resistance visibly decreases with the increasing temperature (Figure R6c), replying enhanced ions transport under relatively high temperature. Above results also suggest that MADG device is able to exert electricity-generating performance under commonly ambient temperature (25°C~40°C). By the way, since the environmental temperature would be usually below 40°C, the electric output of MADG is not further investigated at higher temperature condition.

In essence, the influence of temperature on electricity generation is arisen from affecting water molecules and free ions transport. We set the temperature of around 40°C, which is just served as a proper experimental control condition to systematically investigate the process of moisture adsorption-desorption generation. Actually, the MADG device is able to exert adsorption and desorption enabled power generation under different temperature, making it possible to utilize MADG device in many regions of the earth. The MADG device enables to deliver electric output in the natural condition (Fig. 6i in main text), further demonstrating the practical applicability of this device.

Related discussion of the temperature influence on performance has been added in revised manuscript (Page 13) and Supplementary Information (Page 25).

Figure R5. The voltage-time (a) and current-time profile (b) in moisture adsorption electricity generation under different temperature (~100% RH). The generated voltage and current in moisture adsorption power generation (c) and desorption power generation (d) under different temperature.

Figure R6. (a) Rate of adsorbing water molecules during moisture adsorption under different temperature and ~100% RH. (b) Rate of desorbing water molecules during moisture desorption under different temperature and ~15% RH. (c) The electrochemical impedance spectra of MADG device under different temperature and ~100% RH.

5) Although the applications of MADG are very impressive, comparisons with other devices are always difficult to achieve. On the other hand, many moisture-based power generators have been reporting the voltage and electric current on different resistors connected to the circuit as load. This seems to adequate to this work. Also, authors could report the electric current of MADG through a load resistor uninterruptedly, maybe for 90 hours as is presented in Fig. 6(i).

Reply: The voltage, current, and power density output of MADG with different resistors have been investigated during moisture adsorption-desorption power generation (Figure R7). When the MADG is connected with an external load of 4 kΩ, the voltage, current, and maximum power density can reach 0.19 V, 56 μA, and 120 mW m⁻² during moisture-adsorbing electricity generation. And the MADG connected with a load of 1 kΩ, enables to endow a voltage, current, and maximum power density of 0.2 V, 22 μA, and 48 mW m⁻² in moisture-desorbing power generation.

Furthermore, we have uninterruptedly measured the long-term current output of MADG with a resistor of 6 kΩ in experimental environment (100% RH, 40 °C). Figure R8a indicates the MADG device can effectively offer electric current output for external load. In the outdoor test at Beijing, the generated current of MADG device is able to last for about 90 h under a waving environment of ~25%–90% RH and 15 °C–30 °C (Figure R8b).

As suggested, the corresponding discussion has been supplemented in the revised manuscript (Page 11 and Supplementary Information Page 17).

Figure R7. Voltage and current output of a MADG with varied electric resistances in adsorption (a) and desorption (b) power generation. Output power density as a function with varied electric resistances in adsorption (c) and desorption (d) power generation.

Figure R8. (a) The current output supplied by MADG device connected with a resistor of 6 k Ω . Inset displays the schematic of circuit. (b) Current variation generated by MADG device during outdoor testing (Beijing in China; Time on May 23th~27th, 2021).

Reviewer #3 (Remarks to the Author):

The manuscript by Haiyan Wang et al. proposed a moisture adsorption-desorption power generator (MADG), which demonstrates a cyclic electric output in a dynamic humidity by adsorbing moisture at high RH and desorbing moisture at low RH. Generally, making use of the humidity difference is an interesting approach to address the unsustainable and non-repetitive issues of moisture generators. However, it is not suitable to be claimed as self-maintained, which actually relies heavily on switchable humidity or large humidity difference of environment. Another point of this manuscript is, different from their former power-generating mechanism, the authors proposed a new principle based on ions diffusion driven by ions concentration difference and ion-hydration energy. However, there should be only one major mechanism responsible for these materials with carboxyl functional groups. As to the proposed mechanism, there are still several issues unrevealed especially related to the ion-hydration process. At this current stage, it is not convincing to explain the absorption and desorption process in different humidity. The following issues related to the proposed process should be fully addressed, and direct evidence should be provided.

Reply: We sincerely acknowledge the reviewer for the positive comments and valuably constructive suggestions for further improving our manuscript. As suggested, we have conducted additional experiments and theoretical calculation, as well as added more discussions, especially regarding working mechanism.

As a matter of fact, the previous works on electricity generation induced by single moisture adsorption are mainly based on ion concentration difference as driving force. The all-new electricity generation in current work involves two processes: moisture adsorption and desorption power generation. For moisture adsorption power generation, the process in current work is well consistent with previously reported results, which is also dominated by ion concentration difference. Differing from adsorption process, the moisture desorption power generation was indeed unable to be explained by ion concentration difference induced process. Based on experimental results and theoretical calculations, ion-hydration energy is reasonably proposed as driving force for new process of moisture desorption power generation. Therefore, the current work is an innovative advance on the basis of previous works.

The working processes for moisture adsorption and desorption power generation are carefully discussed as follows (Figure R1). During moisture adsorption process at high RH (Figure R1a), moisture gradually increases from top to bottom in the device along thickness direction, which leads to asymmetrical moisture adsorption and Na^+ ions dissociation in the system, accordingly forming ion concentration difference. It could be seen that barely no hydrated Na^+ ions exist in the lower regions at the initial and middle states of adsorption, as a result diffusion can only drive hydrated Na^+ ions from upper region to lower region driven by ion concentration difference, generating electric signal, which also coincides with previously reported results. At the final state of moisture adsorption, the system will approach to adsorption equilibrium state, water content associated with Na^+ ions should distribute uniformly in the system, leading to disappearance of net ion and current flow.

For moisture desorption process (Figure R1b), the ion concentration of hydrated Na^+ is uniform without ion concentration difference at the beginning state. As water molecules gradually flow out to low RH environment, the water content of device drops. Accordingly, already dissociated free ion would be surrounded by less water molecules in a long range, while it still remains hydrated state rather than direct variation to bonded state (middle state in Figure R1b). In this situation, there is lack of effective ion concentration difference served as driving force in the system. Due to asymmetrically desorption, the water content descends from top to bottom in system. Correspondingly, hydrated Na^+ ions in the upper region are surrounded by fewer water molecules than those in the lower region. By dominated by ion-hydration energy, the hydrated Na^+ ions tend to move from upper region to lower region, where such ions could interact with more water molecules to form a more stable configuration and thus become more energetic favorable. At last state, the moisture desorption would end up with equilibrium. The water content and ion concentration become zero, leading to termination of desorption power generation.

Figure R1. (a) The schematic diagrams of postulated working mechanism of moisture adsorption power generation driven by ion concentration difference. (b) Illustrations of working mechanism of moisture desorption power generation dominated by ion-hydration energy.

Figure R2. (a) ^{23}Na solid NMR spectra of SAG film with different water content during desorption process. (b) Chemical shifts variation with number of water molecules according to first principle calculation. (c) Simulated ion-hydration energy of hydrated Na^+ ion with different numbers of water molecules by MD simulations. Inset are the calculated configurations of hydrated Na^+ ion with 2 or 24 water molecules.

The ion diffusion driven by ion-hydration energy is firstly proposed for moisture desorption power generation, which core relies on two points, including: **(1) Hydrated Na^+ ions number remains unchanged but only surrounding water molecules number reduces with decreasing water content within a wide range, thus there is lack of effective ion concentration difference served as driving force in the system;** **(2) Hydrated Na^+ ion combined with more water molecules would possess lower ion-hydration energy than that of hydrated Na^+ ion surrounded less water molecules.** To verify the first viewpoint, we have conducted solid state nuclear magnetic resonance (NMR) testing to measure the variation of ^{23}Na chemical shift of SAG film with different water content (Figure R2a), and constructed the dependance between ^{23}Na chemical shift and surrounding water molecules number by theoretical calculation (Figure R2b). Given that, taking the chemical shift (viably providing information about microscopic chemical environment) as bridge, the relationship between water molecules number surrounded around ^{23}Na and water content could be constructed reasonably (Figure R2b). As displayed in Figure 2a, two peaks of chemical shift correspond to bonded Na atom anchored on the polymer chains and dissociated Na^+ ions with hydrated state, respectively. The ^{23}Na

chemical shift peak of SAG film with water content of 167% appears ~0 ppm, coinciding with that of completely Na^+ ion of aqueous Na_2SO_4 solution, which suggests the Na totally exists in the form of dissociated hydrated Na^+ ions. The bonded Na stably gives rise to a peak at the chemical shift of 7.2 ppm (marked by black dash line in Figure R2a), while the chemical shift of hydrated Na^+ ion visibly deviated from 0 to 1.7 ppm with decreasing water content from 167% to 20% (marked by green and red dash line) with water content. More interestingly, when the water content gradually decreases from 167% to 50%, NMR spectra always only appear the peak of chemical shift of dissociated Na^+ ion coupled with a gradual positive shift based on shielding effect, demonstrating that Na^+ ion can keep dissociated hydrated state in such wide range (167% ~50%) but surrounding water molecules will be reduced.

Furthermore, based on first principle calculation, the number of water molecules around Na^+ ion corresponding to chemical shift could be quantitative (More calculation details could be found in following Calculation Method). Chemical shifts variation with number of water molecules around Na^+ ion is shown in Figure R2b, where measured chemical shifts at 20%, 38% and 66% water concentrations are marked with dash lines. Combined with theoretical calculation results, the hydrated Na^+ ions will be surrounded by less number of water molecules with reduced water content, reconfirming above first point.

To prove the second point, the hydration energy of hydrated Na^+ ion is calculated by MD simulations (more calculation details could be found in Supplementary Information). As displayed in Figure R2c, the hydrated ion combined with more water molecules would possess the lower ion-hydration energy, attributed to the formation of stable configuration (*J. Phys. Chem.*, 1987, 91, 6269–6271). Therefore, it is reasonably supported that ion-hydration energy could be serve as a driving force.

Figure R3. (a) Schematic illustrations of different directions of moisture source interacted with MADG. (b) Generated electric signals of MADG in response to moisture source in different directions during adsorption and desorption power generation. (c) Illustration of experimental set-up of confocal laser scanning microscope testing. The SAG film infused with fluorescence probe is reasonably packaged, ensuring directional adsorption and desorption of water molecules from one side. (d) Distribution fluorescence intensity along with scanning direction during initial, middle, and final state of adsorption in SAG film, corresponding to the relative distribution of water content in SAG film.

In addition, electric signals generated by adsorption and desorption power generation perform the same direction, which reflects ions diffusion and driving force are in the same direction for these two power generation processes (Figure R3a and b), according with proposed same-directional driving force. During asymmetric water adsorption and desorption, the gradient distribution of water content is also confirmed by confocal laser scanning microscope (CLSM) testing (Figure R3c and d).

In conclusion, we herein develop a self-adaptive power generator, which enable to integrate moisture adsorption power generation driven by ion concentration and desorption power generation dominated by ion-hydration energy. The working mechanism has been demonstrated based on more experiments and theoretical simulation, especially the chemical shift related testing and simulations.

As suggested, more detailed discussions and verifications about working mechanism have been added in the manuscript (Page 9, 10 and 14–17) and highlighted.

(1) The calculation results in Fig. 2e and f are not reasonable. It seems to me that these data were calculated based on the tested curves of Fig.2d, however, most data in different RH states have not reach the equilibrium state, therefore the obtained M_e is not reasonable to be used for the calculation as described in Equation (1) and (2). Also, the authors should explain why the SAG film displays the highest adsorption diffusion coefficient (D) in the low RH of 20% (Fig. 2f). Generally, the SA of porous SAG network that plays main role of water absorption demonstrates much higher water absorption capacity and dynamic in high RH than in low RH at 40°C.

Reply: There could be some misunderstandings in the context. First, the calculations of diffusion coefficient in Fig. 2e and f are not based on the curves of Fig. 2d but are using the kinetic water adsorption and desorption profiles at different RH as shown in Figure R4. The terminations of sorption curves have reached equilibrium states. For the dynamic gravimetric vapor sorption (DVS) test, the RH is increased automatically to the target value and equilibrated by a suitable threshold criterion of a change in mass over time (called d_m/d_t) of greater than 0.002% for 10 consecutive minutes. Therefore, whether it can appear an obvious platform on sorption profiles is correlated with the proportion of adsorbed equilibrium time to total time. Accordingly, there seem to show no platform for some tested adsorption curves, but the adsorption processes of samples have actually reached equilibrium state (*J. Power Sources, 2006, 160, 426–430*). In short, the obtained the mass of water uptake at equilibrium (M_e) for adsorption process fully reasonable to be adopted for calculations. By the way, the sample also have ultimately maintained balanced states under different RH in Figure 2d (Figure R5). Similar kinetics curves (such as Figure R5b) have been reported in other papers (*J. Agric. Food Chem., 2016, 64, 2153–2161*).

Figure R4. Moisture adsorption and desorption profiles under 20% RH (a), 45% RH (b), 70% RH (c) and 99% RH (d) condition (40°C).

Figure R5. (a) Moisture adsorption and desorption kinetics for SAG film at 40°C. (b) Reported water vapor sorption kinetics at 25 °C (*J. Agric. Food Chem.*, 2016, 64, 2153–2161).

Second, the reason of the highest adsorption diffusion coefficient in the low RH are given below. Generally, the moisture adsorption in porous material is mainly composed of the following processes: (1) Water molecules as adsorbate diffuse to the fluid film on material surface of adsorbent (Figure R6a); (2) Water molecules by gas phase diffuse in pores called as pore diffusion, and adsorbed water molecules anchored on pore wall transfer to adjacent sites called as surface diffusion (Figure R6b); (3) Water molecules are adsorbed to the sites on pores (Figure R6c). When water molecules interact with material, they will be preferentially occupied with active sites to form surface coverage. The pore will be filled with water molecules along with continuing surface coverage, which would impede diffusion in pores. The SAG film displays the highest adsorption diffusion coefficient (D) at the low RH of 20%. It could be attributed to sufficient active sites and beneficial diffusion channel without water coverage at low RH. During moisture adsorption at high RH condition, active sites and partial pores have been occupied with water molecules, inducing insufficient sites and unfavorable pores for later adsorption, which results in decreased diffusion coefficient.

Figure R6. Diagram showing moisture adsorption process in porous material.

In addition, the results are further supported by the calculation equation of diffusion coefficient. Based on Fick's second law in the form of a trigonometric series, the initial dynamics of moisture adsorption could be described as:

$$\frac{M_t}{M_e} = \frac{2}{d} \sqrt{\frac{Dt}{\pi}} \quad (0 < \frac{M_t}{M_e} < 0.6)$$

where d , M_t and M_e represents thickness, the mass of water-adsorbing capacity at the time t and equilibrium for adsorption process, respectively. Notably, the applicability of the equation for the calculation of diffusion coefficient need to satisfy the following conditions: (1) $0 < M_t/M_e < 0.6$; (2) The values of M_t/M_e are plotted as a function of square root of time in seconds. Based on above equation and boundary conditions, the diffusion coefficient reflects the rate of water molecules transfer during partial adsorption dynamics ($0 < M_t/M_e < 0.6$), which is not directly relevant to water-adsorbing capacity (M_e). In general, the adsorbed sites and effective pores have influence on diffusion coefficient. It is reasonable that the highest diffusion coefficient remains at low RH, which has been also reported (*J. Power Sources*, 2006, 160, 426–430; *J. Food, Sci.* 1990, 55, 218–231).

As suggested, Figure R4 have been added to Supplementary Fig. 10 and corresponding analysis in the revised manuscript (Page 7 and Supplementary Information page 12).

(2) Based on the theoretical calculation, the authors claimed the ion-hydration energy of ion-water cluster serves as a driving force, which allows the directional ions movement from the moisture-releasing region to well-watered region. As to the proposed ions concentration difference and ion-hydration energy, which driving force is dominant during absorption and desorption process, respectively, and why?

Reply: Thanks for these insightful comments. As presented in the above results and discussions on working mechanism, ion concentration difference is dominant during adsorption process, while for desorption process, ion-hydration energy prevails. The reasons are given below:

In adsorption process (Figure R1a), the moisture gradually increases from top to bottom, and at the end moisture saturates the whole system. It could be seen that barely no Na^+ hydrated ions exist in the lower regions at the initial and middle states, as a result diffusion can only drive hydrated Na^+ ions from upper region to lower region, demonstrating that the driving force of ion concentration difference dominates at this situation.

For desorption process (Figure R1b), the ion concentration of hydrated Na^+ is uniform, ion concentration difference is zero at the beginning state. As water molecules gradually flow out to low RH environment, the water content of device decreases, and already dissociated free ions would be surrounded by less water molecules in a long range, which still remain hydrated state rather than direct variation to bonded state (middle state in Figure R1b). In this situation, there is lack of effective ion concentration difference served as driving force in the system. Correspondingly, hydration energy now dominates and drives hydrated Na^+ ions to transport from upper region to lower region. To be specific,

in upper region, hydrated Na^+ ions are surrounded by less water molecules, and the hydrated Na^+ ions tend to move from upper region to lower region, where they could combine with more water molecules to form a stable configuration and thus become more energetic favorable. The direction of ions flow driven by ion-hydration energy is also consistent with the current signal generated by moisture desorption power generation.

Above discussions of driving force for power generation have been added in revised manuscript (Page 9 and 10) and highlighted

(3) As described in the manuscript, the MADG yields spontaneous ions flow in the same direction during adsorption and desorption process. The induced voltage and current also display the same direction in the high and low RH condition. In this case, does it mean Na^+ ions would finally run out from one side and accumulate at another side? What would happen in the end after cycles? In this case, it is not suitable to be claimed as closed-loop pathway.

Reply: Thanks for your good points. The Na^+ ions would not finally run out from one side and accumulate at the other, and nothing is changed after cycles. Hydrated Na^+ ions can diffuse directionally during ongoing power generation process, that is the middle state of moisture adsorption. Subsequently, no matter for the final state of adsorption or desorption, the system would end up with an equilibrium state, water content associated with Na^+ ions should distribute uniformly in the system without net ions flow. The only difference is that Na^+ ions concentration and water content are nonzero at adsorption final state, while for desorption final state, all of them equal to zero.

In Figure R7, voltage, current and water content evolution curves for adsorption and desorption processed are shown on the left, and the schematic diagrams of typical hydrated Na^+ ions' distribution at given time points (marked by black dash lines in voltage, current and water content evolution curves) are shown on the right. At the final state of adsorption or desorption, the electric signals incline to zero (marked by a3 and d3), which coincides with uniform ions distribution in the system. In addition, sodium element (as shown in energy dispersive spectrum (EDS) results) in the initial state and final state of adsorption mainly uniformly distribute in material (Figure R8), reconfirming no accumulation of Na^+ ions at adsorption or desorption equilibrium.

To see whether it could be claimed as closed-loop pathway, the initial and final states as shown in Figure R7b and d, could provide evidences. It is found that the final state of adsorption prepares the initial state of desorption and vice versa, and therefore, the whole process, i.e. the combination of adsorption and desorption, in this way could circulate. Note that the final state of desorption is same as the adsorption initial state and thus is not given explicitly here.

Related discussion of ions distribution has been added in revised manuscript (Page 9 and 10) and Supplementary Information (Page 9 and 10).

Figure R7 (a) Change in mass (Δm), adsorption rate ($d(\Delta m)/dt$), current and voltage output during moisture adsorption power generation under $\sim 100\%$ RH and 40°C . (b) Schematic diagrams of postulated distribution of hydrated Na^+ ions during adsorption at given time points (marked by black dash lines in (a)). (c) Change in mass (Δm), desorption rate ($-d(\Delta m)/dt$), current and voltage output during moisture desorption power generation under $\sim 15 \pm 5\%$ RH and 40°C . (d) Schematic diagrams of distribution of hydrated Na^+ ions during desorption at given time points (marked by black dash lines in (c)).

Figure R8. (a) Cross-section scanning electron microscopic views of the SAG film. EDS of sodium in the SAG film during initial state of water adsorption (b) and equilibrium state of adsorption (c).

(4) We also noticed that the authors' earlier work on a bilayer film that composed of an analogous material (PSSA) as electricity generator. Actually, as the main part responsible for the electricity generation, PSSA has similar carboxyl functional group with SA in this manuscript, which should share the same electricity-generating principle. However, the authors proposed a different mechanism in these two reports. The authors should clarify it.

Rely: We appreciate the reviewer's careful thought. It should be noted that the proposed mechanism in this work is not contradictory to that in our earlier work. This work is a significant extension on the

basis of previous work. The device based on SA hybrid is able to conduct moisture adsorption power generation driven by ion concentration difference at high RH condition as demonstrated above reply (Figure R1a), which is well consistent with working mechanism of previous works. Furthermore, the ternary composites (SA-SiO₂-rGO, denoted as SAG) possesses robustly crosslinked ionizable network coupled with porous microstructure, thus endowed with desirable water molecules and ions transfer during moisture adsorption and desorption based on synergistic effect, while it is not the case for single PSSA. Interestingly, we further found that the device could also asymmetrically desorb moisture at low RH and accordingly induce power generation, resulted from specific packaging design and unprecedented power-generating materials by SAG. Moreover, differing from adsorption process, we found that the working mechanism of moisture desorption power generation was unable to be explained by ion concentration difference induced process. Therefore, we have tried a lot of experiments and theoretical simulations to investigate the working principle of moisture desorption electricity generation, and ion-hydration energy was proposed as driving force for this process (Figure R1 and R2).

In general, the mechanism of power generation is composed of two parts, including: 1) Moisture adsorption power generation driven by ion concentration difference at high RH, which is matched with previous mechanism; 2) Moisture desorption power generation dominated by ion-hydration energy at low RH. Thus this work is an innovative advance on the basis of earlier work.

As suggested, related discussion has been added in revised manuscript (Page xx) and Supplementary Information (Page 9, 10, and 14–17) and highlighted.

(5) As described in manuscript, the authors claimed the force of ion-hydration energy might surpass that of ions concentration difference and presumably dominate ions migration in low RH condition. While in high RH condition, it was mainly dominated by ions concentration difference. Either Fig.1d or Fig. 3a illustrate the ion-hydration process in the high RH. Therefore, it seems the proposed mechanism is paradox when explain the absorption and desorption process. The author should provide solid evidence.

Reply: As demonstrated in above reply, the ion concentration difference is dominant during moisture adsorption power generation, while for desorption process, ion-hydration energy prevails. According to your suggestion, we have carefully improved Fig. 1d, Fig. 3a and b to clearly diagram the two processes, including ion concentration difference induced process at high RH and ion-hydration energy dominated process at low RH (Figure R9 and R10). We have added the modified diagrams in the revised manuscript (Page 5 and 9).

Figure R9. Scheme of the device structure and working principle of MADG.

Figure R10. Schematic illustrations of the postulated working principle of moisture adsorption power generation (a) and moisture desorption power generation (b).

(6) Actually, there are two kinds of Na^+ ions in the system. One is the dissociated free Na^+ ions, and another is Na ions that still anchored on the polymer chains. The author should make it clear which kinds of the Na^+ ions cline to be hydrated and why? It seems this supposed mechanism only take the dissociated free Na^+ ions into consideration.

Rely: Approvingly, there are dissociated free Na^+ ions and boned Na atom anchored on the polymer chains in the system, as above demonstrated by solid state nuclear magnetic resonance (NMR) testing (Figure R2a). Ions are surrounded by a concentric shell of water in the process of solvation, which could be called ion hydration (*Chemistry-California Edition, Cambell, Boston, 2006, 734, ISBN: 978-0-13-201304-8; https://en.wikipedia.org/wiki/Solvation_shell; Adv. Colloid Interface Sci., 2019, 268, 1–24*). Given that, the dissociated free Na^+ ions surrounded by enough water molecules are already in hydrated state (*Nature 2018, 557, 701–705*), because these ions accomplish bond breaking and hydration processes. On the other hand, every bonded Na atom has nearly same possibility to be

hydrated, since the binding energy between Na^+ and COO^- is the same. The specific surrounding water molecules environment determines whether bonded Na atom is hydrated, and it could be considered as a stochastic process from a macro standpoint. With moisture increasing, bonded Na atoms enables to combine with enough water molecules and overcome the binding energy, stochastically forming hydrated ions.

In our system, only dissociated Na^+ ions could diffuse freely in the system and accordingly induce electric output. The bonded Na atoms may have very small electric dipoles owing to random directions and distribution. Their total influence to the diffusion of dissociated free Na^+ ions is negligible, and the voltage contributed by these dipoles disappear statistically.

Besides, our work only studies the balance voltage (that is the voltage at the steady state, in which all physical quantities become time independent) variation with different dissociated free Na^+ ions concentration, thus many complicated dynamic processes (for instance, anchored Na atoms could be hydrated and become dissociated free Na^+ ions at some points and its reverse process) are unnecessary to take into account.

Based on the above considerations and also to be aware of the major concern in this article is power generation, it is reasonable to consider only dissociated free Na^+ ions in our theoretical calculation and discussion. Related discussion has been added in revised manuscript (Page 14 and 16) and Supplementary Information (Page 3 and 4) and highlighted.

(7) Does this system rely heavy on a high humidity difference (ΔRH)? The author mainly demonstrates the feasibility at extreme condition by changing the RH from $90\pm 10\%$ to $15\pm 5\%$ at $40\text{ }^\circ\text{C}$, that is, induced by a high RH difference ($\Delta\text{RH}=80\%$). However, its universality in other RH condition needs to be verified. For example, whether lower initial RH states (i.e., change from RH= 35% to 15%) or smaller RH difference (i.e., changing from RH=100% to 80%) still work and obtain the same conclusion? Besides, is there any critical RH difference that is required before to achieve sustained performance?

Reply: Thanks for these insightful comments. According to your suggestion, we have examined the power-generating performance and adsorbed or desorbed capacity at different RH. The adsorbed capacity, voltage and current output at different RH in response to moisture adsorption process are shown in Figure R11a and b. In the RH below 20% environment, the MADG device hardly produces electric output owing to very little adsorbed capacity. And the voltage and current outputs are enhanced as the RH increases in the moisture adsorption power generation, because high RH condition is beneficial to water adsorption and improve the uptake amount in device (Figure R11c). As shown in Figure R11d–f, the generated voltage and current outputs raises as the RH decrease in the moisture desorption power generation, arisen from that low RH environment enables to facilitate water desorption in device.

The MADG can generate an voltage of up to 0.5 V and a current of approaching to 100 μA at 100% RH based on adsorption power generation. When the RH changes from 100% to 80%, the hydrated MADG will desorb water molecules and render a voltage of about 0.02 V and current of 3.3 μA . The induced current signal through adsorption power generation is below 1 μA in 35% RH condition due to very little adsorbed capacity. Thus, when the RH changes from 35% to 15%, the electric signal is negligible. The results demonstrate that electric output has no direct dependence on humidity difference.

In addition, when the RH condition above 40% would allow to effectively adsorb water and dissociate free ions, the device will be able to supply electric signals in the adsorption power generation. The device also can perform desorption power generation in response to small variation of humidity (such as 100% to 80%), not relying heaving on a high humidity difference to viably work. Above results coherently demonstrate that the MADG device could exert moisture adsorption and desorption power generation in a broadly workable range of RH.

Related discussion of the RH influence on performance has been added in revised manuscript (Page 13) and Supplementary Information (Page 24).

Figure R11. (a, b) Voltage and current output of device in response to different RH during moisture adsorption power generation (40°C). (c) The mass change of SAG film by water molecule adsorption upon different RH under 40°C. (d, e) Voltage and current output of hydrated device in response to different RH during moisture desorption power generation (40°C). (f) The mass change of saturated SAG film by water molecule adsorption upon different RH under 40°C

(8) Why the temperature of 40 °C was chosen to characterize the moisture adsorption and desorption kinetics and electricity generation performance of SAG film? It has been known that the diffusion dynamic largely related to temperature, how the electricity generation performance at RT? According

to Supplementary Fig. 1, the variation of relative humidity (RH) and temperature is large, the author should take the temperature' effect into consideration.

Reply: According to the valuable suggestions, we have conducted experiments to investigate the effect of temperature on the power-generating performance. This MADG device can generate a voltage output of ~ 0.35 V and a current of ~ 27 μ A at 25°C and 100% high RH, and deliver electric output (~ 0.3 V and ~ 15 μ A) at 25°C and 15% low RH. As shown in Figure R12 a to c, when the value of temperature from 25°C to 40°C (100% RH), the generated voltage of MADG increases 0.53 V, and current output rises from about 82 μ A. It is attributed that relatively high temperature is beneficial to water adsorption and improve the rate of adsorbing water molecules at 100% RH (Figure R13a). Meanwhile, the induced voltage and current output by moisture-desorption process are also enhanced with the temperature from 25°C to 40°C (15% RH, Figure R12d), because proper temperature enables to rise the rate of desorbing water molecules at 15% RH, leading to increasing the desorbing process (Figure R13b). In addition, the electrochemical impedance spectroscopy displays that the ionic resistance visibly decreases with the increasing temperature, replying enhanced ions transport under relatively high temperature (Figure R13c). Above results also suggest that MADG device is able to exert electricity-generating performance under commonly ambient temperature ($25^{\circ}\text{C}\sim 40^{\circ}\text{C}$). By the way, since the environmental temperature would be usually below 40°C , the electric output of MADG is not further investigated at higher temperature condition.

In essence, the influence of temperature on electricity generation is arisen from affecting water molecules and free ions transport and diffusion dynamic. The MADG device is able to exert adsorption and desorption enabled power generation under different temperature. We set the temperature of around 40°C , which is just served as a proper experimental control condition to systematically investigate the process of moisture adsorption-desorption generation.

Related discussion of the temperature influence on performance has been added in revised manuscript (Page 13) and Supplementary Information (Page 25).

Figure R12. The voltage-time (a) and current-time profile (b) in moisture adsorption electricity generation under different temperature (~100% RH). The generated voltage and current in moisture adsorption power generation (c) and desorption power generation (d) under different temperature.

Figure R13. (a) Rate of adsorbing water molecules during moisture adsorption under different temperature and ~100% RH. (b) Rate of desorbing water molecules during moisture desorption under different temperature and ~15% RH. (c) The electrochemical impedance spectra of MADG device under different temperature and ~100% RH.

(9) The authors claimed a universality of the moist-adsorb-desorb-generation; however, Fig.4 f only presents the voltage output for several analogous materials, how about their short-current values?

Reply: As suggested, we have tested the short-circuit current offered other analogous materials (Figure R14). The alginate acid/SiO₂/rGO, PEDOT: PSS/PSSNa, and carrageenan material served as electricity-generating layer, offers the current output of 184, 11, and 10 μA for moisture-adsorbing electricity generation, as well as renders current signal of 165, 1, and 1 for moisture-desorbing power generation (Figure R14a). As shown in Figure R14b, analogous materials with porous and ionizable property, viably exert moisture adsorption-desorption power generation, confirming the universality of working principle.

We have added Figure R14 to Supplementary Fig. 37 (Supplementary Information Page 26) and corresponding analysis and discussion in the main text (Page 14).

Figure R14. (a–c) Short-circuit current generated by alginate acid/SiO₂/rGO (a), PEDOT: PSS/PSSNa (b), and carrageenan film (c) during moisture adsorption power generation. (d–f) Short-circuit current generated by alginate acid/SiO₂/rGO (d), PEDOT: PSS/PSSNa (e), and carrageenan film (f) during moisture desorption electricity generation.

(10) Fig. 3c and d. Is there any method or strategy to control the initial open voltage always start exactly from 0 V?

Reply: Yes. For moisture adsorption generation, when the initial state of device without adsorbed water remains totally dry, the generated initial open-circuit voltage exactly starts from 0 V (Figure R15a), arisen from no ions dissociation and diffusion in device. In order to exhibit a visible platform at 0 V, the testing time can be prolonged in the initial state. For moisture desorption generation, when the initial state of device is in adsorption equilibrium, the induced initial voltage also begins with 0 V owing to no ions concentration difference in device (Figure R15b).

We have added Figure R15 to Supplementary Fig. 18 and corresponding analysis and discussion in the main text (Page 11 and Supplementary Information Page 16).

Figure R15. Voltage-time profiles in moisture adsorption power generation (a) and desorption power generation (b).

(11) According to Fig.4a, the described hierarchical pore structure of SAG film seems play important role to facilitate water molecules and ions transport during the absorption and desorption process, therefore the corresponding BET related to the porosity of SAG and SA-rGO should be provided.

Reply: We conducted the BET analysis through nitrogen sorption to characterize the pore size. Figure R16a displays that the pore size distribution of SAG and SA-rGO is predominately in 2.5 and 1.7 nm, respectively. However, it is worthwhile mentioning that BET measurement is mainly used to analyze nanoscale mesopore (2~50 nm) (*Can. J Chem. Eng.*, 2019, 97, 2781–2791). The SAG and SA-rGO materials both feature with micron-scale pore rather than mesopore. Thus, such BET measurement would be not applicable for investigating the porosity of SAG and SA-rGO.

Furthermore, we adopted a widely accepted method of mercury intrusion porosimetry (MIP) (*J Non-Cryst. Solids*, 2011, 357, 1319–1327) to measure the porosity of SAG and SA-rGO (Figure R16b). The average pore diameter of SAG and SA-rGO remains 2.3 and 1.9 μm . And the porosity of SAG and SA-rGO is 77% and 70%, respectively, demonstrating the SAG film possesses superior porosity. This is consistent with the observation from cross-sectional scanning electron microscopic (SEM) images. The SA-rGO film shows typically lamellar structure (Figure R16f–h). For SAG film, one-dimensional SiO_2 nanofibres are well interconnected with lamellar walls, enabling to forestall layer stacking and form porous network (Figure R16c–e). Accordingly, the SAG film affords the ion conductivity of 0.11 S m^{-1} and water-adsorbing diffusion coefficient of $3.8 \times 10^{-7} \text{ cm}^2 \text{ s}^{-1}$ at 100% RH condition, which are both better than that of values of SA-rGO (0.02 S m^{-1} and $1.0 \times 10^{-7} \text{ cm}^2 \text{ s}^{-1}$), confirming superb water molecules and ions transport in SAG film.

We have added Figure R16 to Supplementary Fig. 26 and 27 (Supplementary Information Page 21) and corresponding analysis and discussion in the main text (Page 13).

Figure R16. Pore size distribution for SAG and SA-rGO film by adopting the BET measurement (a) and MIP testing (b). SEM images at different magnification for SAG film (c–e) and SA-rGO film (f–h).

(12) The performances in Fig. 6l were not consistent with the results obtained in Fig. 3c, d, g and h. For example, Fig. 3c delivers a voltage output (0.15V) after lasted for 3.3 h when maintained in high RH condition (100%), however, Fig. 6l shows the voltage output maintains at ~0.45V after 14.5h during the first period in a relative stable RH condition close to 100%. The author should explain it.

Reply: In fact, the test conditions for Fig. 3c and Fig. 6l are discrepant. The profile of Fig. 3c is examined in an artificially laboratory environment with constant RH and temperature (100% RH, 40°C), which is relatively static and closed state without airflow. But the profile of Fig. 6l is measured in a naturally outdoor condition featured with normal air convection, which is dynamic and open situation. For the first period for 14.5 h in Fig. 6l, the RH and temperature fluctuate at ~90%–100% and ~16°C–21°C. Though the RH conditions seem to be close for Fig. 3c and Fig. 6l, the test conditions differ evidently, such as temperature, convection, and pressure, which viably has an effect on moisture adsorption, ions dissociation and diffusion. Therefore, the performance of Fig. 6l is not exactly consistent with the results in Fig. 3c. And it would be reasonable for the deviation of testing results between outdoor and laboratory environment.

(13) The authors should explain the fluctuation of voltage in Fig. 3g. Besides, why the voltage in Fig. 3h is relative stable while th

Reply: Thanks for your question. The controllable switch point between moisture adsorption and desorption is different between Fig. 3g and Fig. 3h. In the Figure 3g (Figure R17a in reply text), the switch points are selected at the final states of moisture adsorption or desorption. Thus the result displays cyclic and fluctuated electric output. Figure R17b shows the amplification of single pulse-like signal in Fig. 3g. On the other hand, the switch points for Fig. 3h (Figure R17c in reply text) are controlled at middle state of adsorption and desorption, not ultimately equilibrium state. Correspondingly, Fig. 3h in main text replies that the MADG device can also offer a relative stable electric output. Moreover, the influence of different switch state on electric output has been exhibited (Fig. 5e in main text). We have also specified the different conditions for Fig. 3g and 3h to avoid misunderstanding.

As suggested, above related explanation and description have been added in the revised manuscript (Page 9 and 10).

Figure R17. (a) Cyclic current output of repeatable moisture adsorption and desorption power generation for MADG. The blue region represents adsorption process at high RH ($90 \pm 10\%$) and white region denotes desorption process at low RH ($15 \pm 5\%$). The switch point of two processes are selected at final state (that is equilibrium state) of adsorption or desorption. (b) Expanded signal extracted from the plots as marked in (a). (c) Voltage output of MADG device in response to switchable RH. The switch point is selected at middle state of adsorption and desorption.

Calculation Method

First principle calculation for chemical shifts

First principle calculation is directly used to study chemical shifts variation with number of water molecules. Different Na^+ ion is surrounded by different number of H_2O molecules with fixed distance (0.22 nm between O atom and Na^+) and orientation (Figure R18). The more realistic atomic configuration at 300 K for $\text{Na}^+/\text{nH}_2\text{O}$ could be obtained by MD simulations, and the average number (n) of neighbor water molecules within 0.25 nm cutoff is given by statistics (to be specific, 3.91 for $\text{Na}^+/\text{4H}_2\text{O}$, 4.06 for $\text{Na}^+/\text{6H}_2\text{O}$, 4.67 for $\text{Na}^+/\text{12H}_2\text{O}$, and 4.56 for $\text{Na}^+/\text{24H}_2\text{O}$), according to which their chemical shifts could be obtained by interpolating first principle calculation.

Figure R18. Absolute chemical shifts variation with neighbor of water molecule number.

REVIEWER COMMENTS

Reviewer #1 (Remarks to the Author):

In the revised manuscript, I appreciate that the authors do a good job to address all the comments from 3 reviewers. Most of my concerns are well responded. However, I still have two minor comments.

1. I still think that the authors should put the figure of a real device in the main article, not in the supplementary materials. The readers should easily see what a real device look like.

2. The authors should clearly indicate in the "Introduction" that: A MADG unit can generate a high voltage of ~ 0.5 V and a current of $100 \mu\text{A}$ at 100% RH (water adsorption), and deliver electric output (~ 5 V and $\sim 50 \mu\text{A}$) at $15 \pm 5\%$ RH (water desorption).

Reviewer #2 (Remarks to the Author):

The authors have done an excellent job incorporating reviewer feedback to make this manuscript much stronger than it was as submitted earlier. Overall, the range of the claims is appropriate for the evidence provided.

Reviewer #3 (Remarks to the Author):

See the report attached

We appreciated the authors' efforts in providing additional experiments and MD simulations to respond our comments. However, some critical concerns regarding the proposed concept, performance and mechanism have not been fully addressed. Especially, I have some reservation on the newly proposed ion-hydration energy to explain the moisture electricity generation mechanism during desorption. Although I was continued to be impressed by the performance of their device, the manuscript should be revised to make a strong case for its publication in Nature Communication.

1. The design concepts.

Q1. The design is not suitable to be claimed as self-adaptive or self-maintained. Instead, this closed-loop for power generation by adsorption and desorption is highly conditional. It relies heavily on the initial state of adsorption/desorption condition, and large humidity difference between switching condition is required to achieve relative high output.

Q2. Tillandsia is specified for its water adsorption from air at high RH. I am afraid that the statement "transpire water at low RH to maintain its life activities" related with Tillandsia in the manuscript is misleading. It is well known that most water uptake materials would spontaneously adsorb moisture at high RH and desorb moisture at low RH. To some extent, the claimed concept of "bioinspired self-adaptive" is farfetched.

Q3. For a closed-loop moisture generator, what we care about is its feedback and universality when switching between different adsorption/desorption states with varied humidity differences condition. However, the authors did not show the corresponding details in the response to Q7, while just listed the apparent humidity-dependent relationship with performance. According to above discussion and Fig. R11, it appears to us that this device relied heavily on the initial state of adsorption/desorption and large humidity difference between the switching condition to achieve relative high output. If so, it will largely hinder its practical application.

2. The performance.

Q1. As also questioned by the other two reviewers, the authors first showed a performance tested at the control condition in lab at $\sim 100\%$ and $\sim 15\% \pm 5\%$ RH and a high temperature of $40\text{ }^\circ\text{C}$ (Fig. 3 c-d). However, when evaluating the related critical cyclic output of repeatable moisture adsorption/desorption, a switchable condition of RH ($90\pm 10\%$ and $15\pm 5\%$) at $30 \pm 10\text{ }^\circ\text{C}$ was taken (Fig. 3 g-h). It is also confused that both these two test conditions are quite different from outdoor test condition ($\sim 25\%-90\%$ RH and $15-30\text{ }^\circ\text{C}$) (Fig. 7i). Moreover, in other performance evaluations (i.e., results in Fig. 7j), the test temperature condition was not provided. Therefore, it is hard to make a reasonable evaluation on the performance of the design. Also, according to the newly provided results (Fig. R12), the output performances at $25\text{ }^\circ\text{C}$ in a relatively high 100% and low 15% RH are quite low.

Q2. I am wondering whether the moisture in high RH $\sim 100\%$ would condense on the film during these lab tests, especially in the outdoor test condition? If so, whether the condensation of water on device would affect the electricity performance (as Fig. R9c indicated)? The authors should

clarify the given test condition.

3. Materials.

Q1. Although the authors provided the photos of SAG film (Fig.2b and Supplementary Fig. 4), the size of films seems inconsistent with that of SAG in water before and after 5 days (Fig.2c). It is thought that there is quick volume change (swollen) after putting in water. Please clarify it. The authors should compare the size change in horizontal and vertical direction of these samples.

Q2. Based on the fabrication process and high-water uptake property (167%), I am wondering whether the SAG film could also have certain volume change after fully adsorption (swollen) and desorption (shrink). The authors should provide the photo and accurate size of the SAG sample at initial state, after adsorption (100% RH, 40 °C), and after desorption (15%RH, 40 °C). If there is some size change, especially in the vertical direction, the working mechanism and output performance will be largely affected.

4. Mechanism.

My biggest concern lies in the driving force for the Na⁺ ion motion in the desorption. For example, it is hard to understand why the device delivers similar voltage (~0.5 V) with different current short-circuit during the absorption and desorption (Fig. 3c-d). It is also difficult to understand why the two driving forces claimed by the authors are at the same order of magnitude. Although the authors used the MD simulation to validate their points, a direct evidence is necessary to justify this critical important point.

More importantly, even all these supposed mechanisms are right, the Na⁺ of SAG system would finally run out from upper side and accumulate at low side after cycles due to the spontaneous Na⁺ ions flow in the same direction (from the upper side to the lower side) both in the adsorption and desorption process. I am afraid that the mechanism demonstrated in the response (Fig. 3a-b or Fig. R1) is questionable. When illustrated the mechanism process, the authors only showed the dynamic process (initial, diffusion, and equilibrium state) of the hydrated Na⁺ ions, whereas without considering the overall Na⁺ ions in SAG system. How about the evolution of the bonded

Na⁺ on SA chains during the adsorption and desorption process?

According to Fig. 5a, the NMR result of SAG with 167% water content only demonstrates the peak of dissociated hydrated Na⁺, suggesting that there is no bonded Na⁺ in the system. In this case, I am puzzled that how could SAG get the equilibrium state as illustrated in Fig. R1a.

Point-by-Point Responses to the Reviewers' Comments

Reviewer #1 (Remarks to the Author):

In the revised manuscript, I appreciate that the authors do a good job to address all the comments from 3 reviewers. Most of my concerns are well responded. However, I still have two minor comments.

Reply: We greatly acknowledge the reviewer for the positive recommendation and are grateful for the helpful and constructive comments to the manuscript.

1. I still think that the authors should put the figure of a real device in the main article, not in the supplementary materials. The readers should easily see what a real device look like.

Reply: Thanks for your good points. As you suggested, we put the figure of a real device into Fig. 1 in the revised manuscript (Page 4).

2. The authors should clearly indicate in the “Introduction” that: A MADG unit can generate a high voltage of ~ 0.5 V and a current of $100 \mu\text{A}$ at 100% RH (water adsorption), and deliver electric output (~ 0.5 V and $\sim 50 \mu\text{A}$) at $15 \pm 5\%$ RH (water desorption).

Reply: According to your valuable suggestion, above sentences have been added into “Introduction” and highlighted in the revised manuscript (Page 4).

Reviewer #2 (Remarks to the Author):

The authors have done an excellent job incorporating reviewer feedback to make this manuscript much stronger than it was as submitted earlier. Overall, the range of the claims is appropriate for the evidence provided.

Reply: We really appreciate the referee for the comments.

Reviewer #3 (Remarks to the Author):

We appreciated the authors' efforts in providing additional experiments and MD simulations to respond our comments. However, some critical concerns regarding the proposed concept, performance and mechanism have not been fully addressed. Especially, I have some reservation on the newly proposed ion-hydration energy to explain the moisture electricity generation mechanism during desorption. Although I was continued to be impressed by the performance of their device, the manuscript should be revised to make a strong case for its publication in Nature Communication.

Reply: We greatly thank the referee for the positive comments and constructive suggestions for improving our manuscript. Accordingly, we have conducted additional experiments and added more discussions in the revised manuscript. Some modifications and explanations are summarized as below.

1. The design concepts.

Q1. The design is not suitable to be claimed as self-adaptive or self-maintained. Instead, this closed-loop for power generation by adsorption and desorption is highly conditional. It relies heavily on the initial state of adsorption/desorption condition, and large humidity difference between switching condition is required to achieve relative high output.

Reply: Thanks for your good points. We have removed “self-adaptive” and “self-maintained” descriptions, as well as carefully improve the elaborations of the design concept in the revised manuscript to ensure the article rigor (Page 1–5, 11 and 22).

Approvingly, the initial state and humidity difference have direct influence on moisture adsorption and desorption, thus the electric output of MADG device is associated with initial state and humidity difference. It should be pointed out that environmental humidity is indeed changing dynamically. In the case, the device is designed to harness such phenomena, which enables to spontaneously adsorb or desorb moisture in response to humidity variation, leading to electric output.

Figure R1. (a) Voltage, current, and mass change of device at different ΔRH during moisture adsorption power generation ($40^\circ C$). The initial state of device remains dry and the RH could

be considered as $\sim 0\%$. (b) Voltage, current, and mass change of device at different ΔRH during moisture desorption power generation (40°C). The initial state is $\sim 100\%$ RH.

Furthermore, we have done control experiments to investigate the effect of relative humidity variation (ΔRH) on electric output (Figure R1). Expectedly, when the initial state of device maintains constant, the electric output is enhanced with increased ΔRH . In addition, we have tested the performance of device in the natural environment. The device enables to deliver electric output in the natural condition (Figure R2), verifying that the device possesses certain environment-adaptive ability.

Figure R2. Voltage output of device during outdoor testing in varied humidity and temperature condition.

Q2. Tillandsia is specified for its water adsorption from air at high RH. I am afraid that the statement “transpire water at low RH to maintain its life activities” related with Tillandsia in the manuscript is misleading. It is well known that most water uptake materials would spontaneously adsorb moisture at high RH and desorb moisture at low RH. To some extent, the claimed concept of “bioinspired self-adaptive” is farfetched.

Reply: We agree with this insightful point. The moisture adsorption and desorption are indeed a common phenomenon in the nature. Based on this phenomenon, we developed moisture adsorption-desorption power generator. In order to precisely elaborate our design concepts and viewpoints, the “bioinspired self-adaptive” words have been removed, and related discussions also have been modified in the revised manuscript (Page 1–5).

Q3. For a closed-loop moisture generator, what we care about is its feedback and universality when switching between different adsorption/desorption states with varied humidity differences condition. However, the authors did not show the corresponding details in the response to Q7, while just listed the apparent humidity-dependent relationship with performance. According to above discussion and Fig. R11, it appears to us that this device relied heavily on the initial state of adsorption/desorption and large humidity difference between the switching condition to achieve relative high output. If so, it will largely hinder its practical application.

Reply: As suggested, more corresponding details have been supplemented in the response to previous Q7. For moisture adsorption power generation, the initial state of device remains dry and the RH could be considered as ~0%. Then the electric output of device is examined at different humidity conditions. As shown in Figure R1a, the generated voltage, current, and moisture- adsorbed amount are increased with Δ RH from ~20% to 100%. Meanwhile, the adsorption-saturated device at 100% RH is used to investigate the electric output in the moisture desorption power generation. As displayed in Figure R1b, the induced voltage, current, and moisture- desorbed amount are also enhanced with Δ RH from ~20% to 90% in the moisture desorption power generation. These results suggest that the electric output of device is related to Δ RH, according with above points of reviewer.

To further demonstrate practical application, we have carried out the outdoor test of devices in varied humidity following environmental changes. The device enables to deliver considerable electric output in the dynamic environment (Figure R2), suggesting its practical applicability.

In the revised manuscript, we have added related details as suggested (Supplementary Information Page 24).

2. The performance.

Q1. As also questioned by the other two reviewers, the authors first showed a performance tested at the control condition in lab at ~100% and ~15% \pm 5% RH and a high temperature of 40 °C (Fig. 3 c-d). However, when evaluating the related critical cyclic output of repeatable moisture adsorption/desorption, a switchable condition of RH (90 \pm 10% and 15 \pm 5%) at 30 \pm 10 °C was taken (Fig. 3 g-h). It is also confused that both these two test conditions are quite different from outdoor test condition (~25%-90% RH and 15-30 °C) (Fig. 7i). Moreover, in other performance evaluations (i.e., results in Fig. 7j), the test temperature condition was not provided. Therefore, it is hard to make a reasonable evaluation on the performance of the design. Also, according to the newly provided results (Fig. R12), the output performances at 25 °C in a relatively high 100% and low 15% RH are quite low.

Reply: Thanks for your good comments. We further clarified the logical context of experiments presented therein. **First**, “~100% and ~15 \pm 5% RH” is control condition in lab. By means of this condition, we explored the electricity-generating performance of device at varied temperature from 25°C to 40°C, and systematically demonstrated the moisture adsorption and desorption power generation at 40°C. **Second**, the natural environment usually changes dynamically, and the humidity and temperature are fluctuating. Given that the fluctuation of ambient environment, we further simulated a dynamic ambient condition of in lab to examine the electricity-generating performance and output continuity of device under continuously dynamic environment. Thus, “90 \pm 10% and 15 \pm 5% RH and 30 \pm 10°C” is served as a simulated

ambient condition. **Third**, in order to demonstrate the potential applicability of device in practical environment, we also tested the electric output of the device at outdoors (Beijing in China, Time on May 23th~27th, 2021). The humidity and temperature were recorded in real-time through sensor during outdoor test. The practical measured humidity and temperature fluctuated in the range of ~25%–90% RH and 15 °C–30°C. And the application demonstration (Fig. 7j) was carried out in the same outdoors. We also supplemented related descriptions of test conditions. In addition, we further carefully improved related elaborations in the revised manuscript (Page 8, 11 and 21)

Q2. I am wondering whether the moisture in high RH ~100% would condense on the film during these lab tests, especially in the outdoor test condition? If so, whether the condensation of water on device would affect the electricity performance (as Fig. R9c indicated)? The authors should clarify the given test condition.

Reply: In the laboratory tests, the device surface has no visible appearance of water condensation under high RH (~100%) condition, and surrounding measurement box surface has a little condensed water. Also, the device surface has no visible appearance of water condensation in the outdoor tests. To reply the first reviewer's question on generating duration time of saturated device, we investigated the electric output of saturated device by directly dropping water into device (as previous Fig. R9c indicated). For this test condition, the device is in atmospheric environment (~20±5% RH and ~25°C), and a few drops of water (~200 μL) are periodically added into device. According to your suggestion, we could systematically investigate the influence of water condensation on moisture power generation in future studies.

3. Materials.

Q1. Although the authors provided the photos of SAG film (Fig.2b and Supplementary Fig. 4), the size of films seems inconsistent with that of SAG in water before and after 5 days (Fig.2c). It is thought that there is quick volume change (swollen) after putting in water. Please clarify it. The authors should compare the size change in horizontal and vertical direction of these samples.

Reply: In fact, we immersed the SAG film completely into water to demonstrate its water-resistance and stability (Fig. 2c in the main text). However, in our device, the SAG film is

encapsulated and fixed by resin. The encapsulation layer could effectively exert confinement for SAG film. On the other hand, the test condition was at 100% RH environment, rather than immersed in water. Therefore, the SAG film has no visible volume change in the device.

Q2. Based on the fabrication process and high-water uptake property (167%), I am wondering whether the SAG film could also have certain volume change after fully adsorption (swollen) and desorption (shrink). The authors should provide the photo and accurate size of the SAG sample at initial state, after adsorption (100% RH, 40 °C), and after desorption (15%RH, 40 °C). If there is some size change, especially in the vertical direction, the working mechanism and output performance will be largely affected.

Reply: Thanks for your suggestion. The SAG film is encapsulated and fixed in our device. As suggested, we have supplemented the photos of SAG film at initial state, after adsorption, and after desorption (Figure R3). There is no visible volume change of SAG film after moisture adsorption and desorption. It could be attributed to effective confinement function of resin encapsulation layer. Related discussion has been added in revised manuscript (Page 9).

Figure R3. The photos of SAG film at initial state (a), after adsorption (b), and after desorption (c).

4. Mechanism.

We thank the reviewer for the questions about underlying mechanism. To clearly respond to all the points, we have divided questions into four parts to reply, and reversed the order of reply in Q4-3 and Q4-2.

Q4-1 My biggest concern lies in the driving force for the Na^+ ion motion in the desorption. For example, it is hard to understand why the device delivers similar voltage (~ 0.5 V) with different current short-circuit during the absorption and desorption (Fig. 3c-d). It is also difficult to understand why the two driving forces claimed by the authors are at the same order of magnitude. Although the authors used the MD simulation to validate their points, a direct evidence is necessary to justify this critical important point.

Reply: Thanks for your good points. Based on experimental and theoretical results, we have conducted comprehensive analyses for mechanism related to the verification of ion-hydration,

magnitude of two driving force, and the explanation of different short-circuit current.

1. Verification of ion-hydration energy based on the both experimental and MD simulation (Figure R4).

Figure R4. Research route on verification of ion-hydration energy.

(1) Electric signals in the same direction. In our study, the device can asymmetrically adsorb and desorb moisture, leading to electric output. For moisture adsorption power generation, such process is dominated by ion concentration difference, according with previously reported results. However, when we repeatedly validated the performance of moisture desorption power generation, we found a relatively abnormal phenomenon. Electric signals generated by adsorption and desorption power generation perform the same direction (Figure R5), demonstrating the directions of ions diffusion are the same for two processes, which cannot be elucidated by the single driving force of ion concentration difference (Figure R6). As shown in Figure R6b, if the desorption power generation is also dominated by ion concentration difference, the generated ions flow and current signal should be contrary to that of adsorption power generation. **Hence, there is certainly other driving force that has the same direction with the water content gradient for the all-new desorption power generation (Figure R6c).**

Figure R5. (a) Generated electric signals of device in response to moisture source in different directions during adsorption (solid line) and desorption power generation (dotted line). Insets

are schematic illustrations of different directions of moisture source interacted with device. (b) The current-time profile in the adsorption-desorption power generation with various adsorption time.

Figure R6. (a) The schematic diagrams of adsorption power generation dominated by ion concentration difference. (b, c) Illustrations of desorption power generation dominated by hypothetical ion concentration difference (b) or other driving force (c).

(2) NMR testing. To further explore the origin of the driving force, we adopted NMR testing to analyze the chemical environment around Na^+ ions during desorption process. More interestingly, with the decreasing water content from 167% to 50%, NMR spectra of SAG film display gradually positive shift and maintain one peak (Figure R7), **solidly identifying the dissociated hydrated Na^+ ions possess different chemical environments corresponding to varied water content.** And varied chemical environment will lead to different ion-hydration energy. We reasonably inferred that other driving force would be ion-hydration energy.

Figure R7. ^{23}Na NMR spectra of SAG film with different water content during desorption.

(3) MD simulations. According to MD simulations, we found that the hydrated ion combined with more water molecules could possess the lower ion-hydration energy (Figure R8). Thus, it is reasonably supported that **ion-hydration energy could serve as a driving force**.

Figure R8. (a) Chemical shifts variation with number of water molecules according to first principle calculation. (b) Simulated ion-hydration energy of hydrated Na^+ ion with different numbers of water molecules by MD simulation.

(4) Consistence. When the proposed ion-hydration energy is served as driving force for desorption power generation, corresponding current direction is consistent with experimental results. Additional evidences (including visualization of water content distribution, related electric results, and kinetic Monte Carlo simulation) coherently reconfirm the validity of proposed driving force.

2. Two driving forces with the close order of magnitude

As two driving forces have different origins, a rigorous magnitude comparison is not possible since their application objects are different. Fortunately, the generated voltage (the electrical driving force to electrons) is a good indicator to roughly compare the magnitude of driving force (Pearson Education, Knight, Randall D., 2004, *Physics for Scientists and Engineers: A Strategic Approach*. 879–995). According to the similar voltage outputs measured by experiments, we would infer that two driving forces (ion concentration difference and ion-hydration energy) lie in close order of magnitude. Moreover, two driving forces both originate from water content gradient in essence. And the water content gradient is very close during adsorption and desorption processes by visualization of water content distribution (see Fig. 6f in the main text), coinciding with above deduction.

3. Different short-circuit current

Both ions concentration and water content of device are very different at most of the time during adsorption and desorption, suggesting varied ion mobility, which may lead to different internal resistances from macroscopic perspective. From Ohm law, different short-circuit current could be expected.

In conclusion, the proposed driving force of ion-hydration energy is reasonable and convincing. The proposed moisture desorption power generation and driving force of ion-hydration energy, provide an all-new pathway and principle for moisture enable power generation, which thus presents the novelty and significance of this work. In our future research, we will further study microscopic and interfacial-interaction mechanisms, such as quantification of ion-hydration energy, construction mathematical model, and universality verification. And we have added related elaborations of proposed driving force to improve logicity in the revised manuscript (Page 17 and 22).

Q4-3 When illustrated the mechanism process, the authors only showed the dynamic process (initial, diffusion, and equilibrium state) of the hydrated Na^+ ions, whereas without considering the overall Na^+ ions in SAG system. How about the evolution of the bonded Na^+ on SA chains during the adsorption and desorption process?

Figure R9. (a) Schematic of the formation of hydrated Na^+ ion from NaCl crystal. (b) Molecular structure of SA. (c) Scheme of the formation of hydrated Na^+ from SA.

Reply: It is a good point. There could be some difference in understanding of the “ion” concept. In fact, salts will dissolve in the water and the water molecules are bounded with the dissociated ions (Figure R9a), spontaneously forming hydrated ions (e.g., $[\text{Na}(\text{H}_2\text{O})_n]^+$). As displayed in Figure R9a, **the dissociated sodium ions are all in the form of hydrated sodium ions**, which are denoted as Na^+ ions to be easy described. The bonded Na atoms is confined in the crystal framework and thus difficult to free migration, which are not generally to be denoted as Na^+ ions. Analogously, sodium alginate (SA) of SAG film can form hydrated Na^+ ions based on the interaction between SA and water molecules (Figure R9b and c). Unlike Cl^- ion of sodium chloride, counter ion ($-\text{COO}^-$) of SA is difficult for free diffusion, since it possesses polymer chains. **The bonded Na atoms of SA is also confined strictly, which are not considered as Na^+ ions.**

More importantly, the directional transfer of charges is the basis of electricity generation for many devices, such as photovoltaic, thermoelectric and moist-electric (*Chem. Rev.*, 2010, 110, 11, 6689–6735; *Science*, 2020, 368, 1091–1098; *Adv. Mater.*, 2020, 32, 2003722). Given that, the bonded Na is not allowed to diffuse, thus causing hardly contributes nothing for moisture power generation. It is known that hydrated Na⁺ ions enable to migrate freely. Therefore, we discuss the migration of hydrated Na⁺ ions (that is overall Na⁺ ions) excluding that of bonded Na atoms in process of power generation, according with other reported power generation studies based on ions diffusion (*Nature*, 2020, 578, 550–554; *Adv. Funct. Mater.*, 2021, 2011016).

Q4-2 More importantly, even all these supposed mechanisms are right, the Na⁺ of SAG system would finally run out from upper side and accumulate at low side after cycles due to the spontaneous Na⁺ ions flow in the same direction (from the upper side to the lower side) both in the adsorption and desorption process. I am afraid that the mechanism demonstrated in the response (Fig. 3a-b or Fig. R1) is questionable.

Reply: There could be some misunderstandings in the context. We have further clarified specific discussions on mechanism and ions diffusion in the revised manuscript (Page 9). At the final state, the Na⁺ ions would not run out from one side and accumulate at the other (Figure R10). Actually, there is Na⁺ ions flow in the same direction both in the adsorption and desorption power generation, coinciding with that the induced current signals possess the same direction. The ions diffusion in the middle states could not necessarily imply ions accumulation in the final states. Obviously, **the assumption of ions accumulation in the final state is entirely contrary to nearly zero-voltage and zero-current outputs** (marked by dotted line in Figure R10a and c), because ions accumulation could cause the formation of electric potential according to Nernst equation. And the zero-outputs also imply the uniform distribution of ions in the final state arisen from the disappearance of asymmetric water distribution.

Figure R10. (a) Change in mass (Δm), adsorption rate ($d(\Delta m)/dt$), current and voltage variation during moisture adsorption power generation. (b) Schematic of postulated working mechanism for adsorption power generation. (c) Change in mass (Δm), desorption rate ($-d(\Delta m)/dt$), current and voltage output during desorption power generation. (d) Scheme of postulated working mechanism for desorption power generation.

For moisture adsorption power generation, the final state refers to moisture adsorption approaching to saturation. Notably, from middle state to final state (b2 and b3 in Figure R10b), the device continuously adsorbs moisture. With the adsorption saturation of device, the distribution of water content in the device keeps uniform, leading to uniform ions distribution. Therefore, though ions directionally migrate in the middle state, ions still tend to be well-distributed in the final state due to uniform distribution of water content. The generated electric signals incline to zero at the final state (marked by dotted line in Figure R10a), confirming uniform ions distribution in the system. **For moisture desorption power generation**, the final state corresponds to desorption equilibrium of device with uniform water distribution. Extremely, when the water content of device tends to zero with desorption, the dissociated Na^+ ions will be entirely transformed into bonded Na atoms driven by electrostatic interaction. As shown in Figure R10c, the zero-outputs (marked by dotted line) also imply uniform ions distribution at the final state of desorption power generation. Overall, above results and analyses coherently reconfirm no ions accumulation at the final state, and reflect the rationality of proposed working mechanism.

Q4-4 According to Fig. 5a, the NMR result of SAG with 167% water content only demonstrates the peak of dissociated hydrated Na^+ , suggesting that there is no bonded Na^+ in the system. In this case, I am puzzled that how could SAG get the equilibrium state as illustrated in Fig. R1a.

Figure R11. ^{23}Na solid state NMR spectra of Na_2SO_4 solution and SAG film.

Reply: Thanks for the comments. The ^{23}Na chemical shift peak of SAG film with water content of 167% appears ~ 0 ppm (Fig. 5a in the main text). As shown in Figure R11, the NMR result of Na_2SO_4 solution also displays a peak with chemical shift of 0 ppm. So, the form of Na in the SAG film with 167% water content is reasonably presumed as dissociated hydrated Na^+ ions. On the other hand, the bonded Na corresponds to a peak at the chemical shift of 7.2 ppm according to the NMR result of dry SAG film. Therefore, the SAG film with 167% water content possesses hydrated Na^+ ions without bonded Na (the meaning of hydrated Na^+ ions and bonded Na in this article are explained in above Q4-3 question).

In addition, the equilibrium state Fig. R1a refers to moisture adsorption approaching to saturation in device, that is final state of moisture power generation. As demonstrated above reply Q4-2, due to uniform distribution of water content at final state, dissociated hydrated Na^+ ions correspondingly tend to be uniformly distributed.

REVIEWERS' COMMENTS

Reviewer #3 (Remarks to the Author):

The authors had made proper revision in addressing our comments. Although I have some reservation on the mechanism responsible for the current flow of the same direction in the adsorption and desorption, the work can be first published. I suggest the authors to try the electrode with the same size and morphology. My feeling is that the electrode plays a main role for the electricity generation.

Point-by-Point Responses to the Reviewers' Comments

Reviewer #3 (Remarks to the Author):

The authors had made proper revision in addressing our comments. Although I have some reservation on the mechanism responsible for the current flow of the same direction in the adsorption and desorption, the work can be first published. I suggest the authors to try the electrode with the same size and morphology. My feeling is that the electrode plays a main role for the electricity generation.

Reply: We really appreciate the referee for the recommending acceptance of our manuscript. In fact, asymmetric moisture stimuli play a significant role in electricity generation rather than electrode. As suggested, we also have tried electrode with same size and morphology and tested the power-generating performance of device. As shown in Figure R1a and b, the device is assembled by the same gold electrode with holes and encapsulated with resin, which enables to allow one-way moisture stimuli and induce water content gradient in the device. Correspondingly, such device could generate electric signal output (Figure R1d and e). And the direction of electric signal is related to that of moisture stimuli, demonstrating that the power generation of device originates from moisture stimuli. As displayed in Figure R1c and f, the device without encapsulation layer is unable to construct water content gradient due to symmetric moisture stimuli, leading to electric signal loss. Therefore, above results are consistent with proposed mechanism in our work.

Figure R1. (a–c) Schematic illustrations of device interacted with different directions of moisture source. The device is assembled by the same gold electrode with holes. (d–f) Generated electric signals of device with the same electrode in response to moisture source in different directions.